# Building electrode skins for ultra-stable potassium metal batteries

Hongbo Ding[1], Jue Wang[2], Jiang Zhou [3], Chengxin Wang [4] ✉ & Bingan Lu [1] ✉

In nature, the human body is a perfect self-organizing and self-repairing system, with the skin protecting the internal organs and tissues from external damages. In this work, inspired by the human skin, we design a metal electrode skin (MES) to protect the metal interface. MES can increase the flatness of electrode and uniform the electric field distribution, inhibiting the growth of dendrites. In detail, an artificial film made of fluorinated graphene oxide serves as the first protection layer. At molecular level, fluorine is released and in-situ formed a robust SEI as the second protection "skin" for metal anode. As a result, Cu@MES||K asymmetric cell is able to achieve an unprecedented cycle life (over 1600 cycles). More impressively, the full cell of K@MES||Prussian blue exhibits a long cycle lifespan over 5000 cycles. This work illustrates a mechanism for metal electrode protection and provides a strategy for the applying bionics in batteries.

Compared with traditional anode materials, metal anodes present the lower redox potential and higher specific capacity, which are ideal anode materials for high energy density batteries[1–13]. However, metal anodes display an extremely high reactivity and easily form an unstable solid electrolyte interface (SEI) after contacting electrolytes[14]. The metal electrode with an unstable interface is unable to control the morphology of the plating/stripping during the charging and discharging process, resulting in the growth of metal dendrites[15,16]. Meanwhile, the volume change caused by the uneven diffusion and deposition of metal ions can exacerbate the growth of dendrites. These issues will cause the rupture of the interfacial SEI and the loss of active species[17]. Therefore, there are still huge obstacles for the practical use of metal batteries[18,19].

In recent years, in-depth research has been carried out on designing metal electrode and formulating electrolyte to inhibit the dendrite growth[20]. For example, constructing a three-dimensional framework to alleviate the volume change and promote uniform ion flux of metal electrodes[21], or designing liquid alloys as metal anodes to suppress dendrite growth have been explored[22]. These methods can address some issues caused by metal electrodes to a certain extent, but they are still far from large-scale applications. Recent studies have shown that the stable interface, which is closely related to SEI, plays a crucial role in the stable cycling of metal batteries[23]. For metal anode, the SEI acts as an ionic conductor between electrode and electrolyte, which effectively avoids the continuous side reactions and regulates the metal nucleation[24]. However, the naturally formed SEI has poor mechanical properties and cannot adapt to volume changes during plating/stripping, causing the rupture of the SEI which exacerbates the growth of dendrites[25]. Researchers have developed a self-catalyzed tribo-electrochemistry strategy to construct a continuous and compact protective layer on the K electrode surface. This continuous and compact protective layer not only improves $K^+$ diffusion dynamics, but also inhibits K dendrite formation by increasing $K^+$ conductivity and decreasing electron conductivity with the amorphous KF. However, this strategy experienced problems in the cycle life and large-scale preparation. The constantly exposed fresh metal surface continuously generates a new SEI, resulting in the continuous consumption of electrolyte and metal. The instability of the interface leads to a rapid decay of the Coulombic efficiency (CE), reducing the rate capability of metal batteries. To solve these problems,

[1]School of Physics and Electronics, State Key Laboratory of Advanced Design and Manufacturing for Vehicle Body, Hunan University, Changsha 410082, China. [2]College of Chemistry and Chemical Engineering, Central South University, Changsha 410083, China. [3]School of Materials Science and Engineering and Key Laboratory of Nonferrous Metal Materials Science and Engineering, Ministry of Education, Central South University, Changsha 410083, China. [4]State Key Laboratory of Optoelectronic Materials and Technologies, School of Materials Science and Engineering, Sun Yat-sen (Zhongshan) University, Guangzhou 510275, China. ✉e-mail: wchengx@mail.sysu.edu.cn; luba2012@hnu.edu.cn

researchers have carried out study in many aspects[26], such as changing the solvent or solvation structure, adding functionalized electrolyte additives, and changing the electrolyte concentration. These methods can build relatively stable interfaces on the metal surface and inhibit dendrite growth of the initial cycle[27,28]. However, the SEI formed only by the optimization of the electrolyte does not yield an excellent mechanical stability, and the scalability and cost are not suitable for the large-scale production[29,30].

Skin is the largest organ of the human body, covering the surface of body with a direct contact to the external environment[31]. The skin is divided into two layers, the epidermis and dermis. Epidermis is on the surface of skin, mainly functioning as a barrier, to defend against the mechanical damage, physical damage, chemical damage, and microorganisms. The role of dermis is to make the skin more extensible and elastic, which protects the capillaries, glands and various substances in the dermal tissue. There are many ion channels in the dermis that can be used for absorption of water and nutrients[32]. This unique structure and properties have important implications on the fabrication of metallic batteries.

Here we show a strategy for stabilizing the metal interface with a metal electrode skin (MES) mimicking human skin. The F-GO is transferred to Cu foil using a simple and effective process, and then relocated to the metal surface using a rolling process. MES has a high surface flatness, which controls the surface electric field intensity during the initial cycle. A uniform electric field affects the ion concentration, allowing ions to deposit uniformly on the electrode

surface. In the process of initial plating, F-GO accommodates some metal and greatly alleviates the volume change of the interface. Alkali metals can promote carbon-fluorine bond cleavage to generate fluoride-rich SEI, which enhances the mechanical strength of SEI. The enhanced interface passivates the highly active metal anode and improves the interfacial stability during plating/stripping. Therefore, under a current density of 0.5 mA cm$^{-2}$ and capacity of 0.5 mAh cm$^{-2}$, the plating/stripping life of the symmetric cell is as long as 2300 h for the metal anode covered with the MES. The Cu@MES||K asymmetric cells reaches a cycle life more than 3200 h (1600 cycles), the longest cycle life of the asymmetric cell to date. This proves that MES can greatly improve the interfacial stability of electrodes. Paired with a Prussian blue (PB) cathode, the full battery exhibits the excellent cycle stability and rate capability (1000 mA g$^{-1}$, cycle lifespan over 5000 cycles). The attractive properties and stability demonstrate the great potential of MES for high-energy metal batteries, which may pave the way for next-generation battery-driven applications.

## Results

### Design and structure of MES

Potassium metal battery (PMB) is identified as one of the most promising candidates for next-generation energy storage devices due to the low redox potential and high theoretical capacity of potassium (K) metal. Compared to published anode materials for K-ion battery, K metal can provide a higher energy density (Fig. 1a)[1–13]. When the battery is charged at a fixed current density, the flux of K$^+$ is more

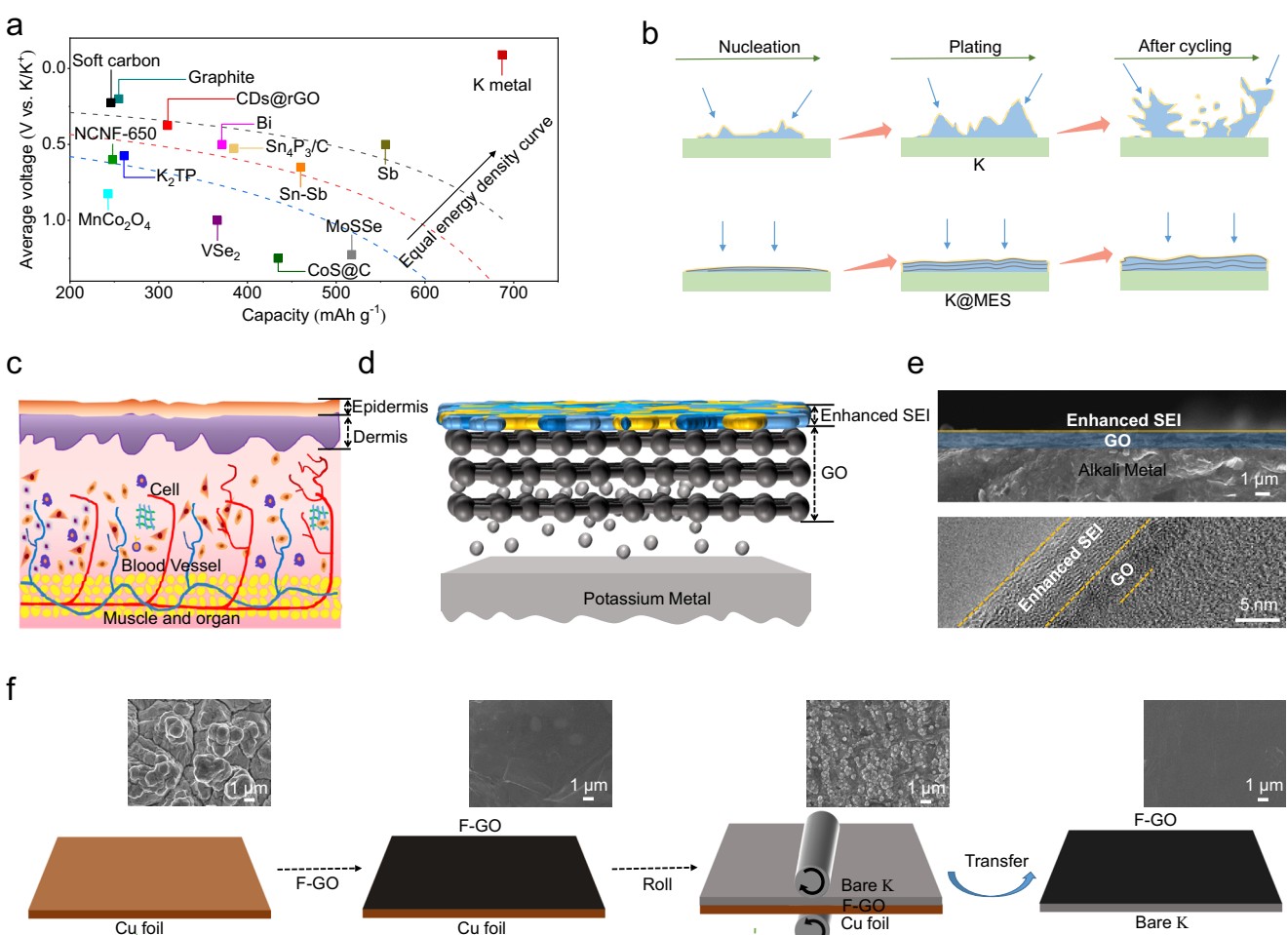

**Fig. 1 | Advantages of K metal cell and preparation of electrode skin. a** The summarized capacity-voltage plots of different anode materials. **b** Schematic illustration of electrodeposition behaviors of K and K@MES. **c** Schematic diagram of human skin structure and surface. **d** Schematic diagram of MES structure and surface. **e** SEM images of K@MES and TEM image of enhanced SEI. **f** Preparation and transfer of MES host material and SEM image.

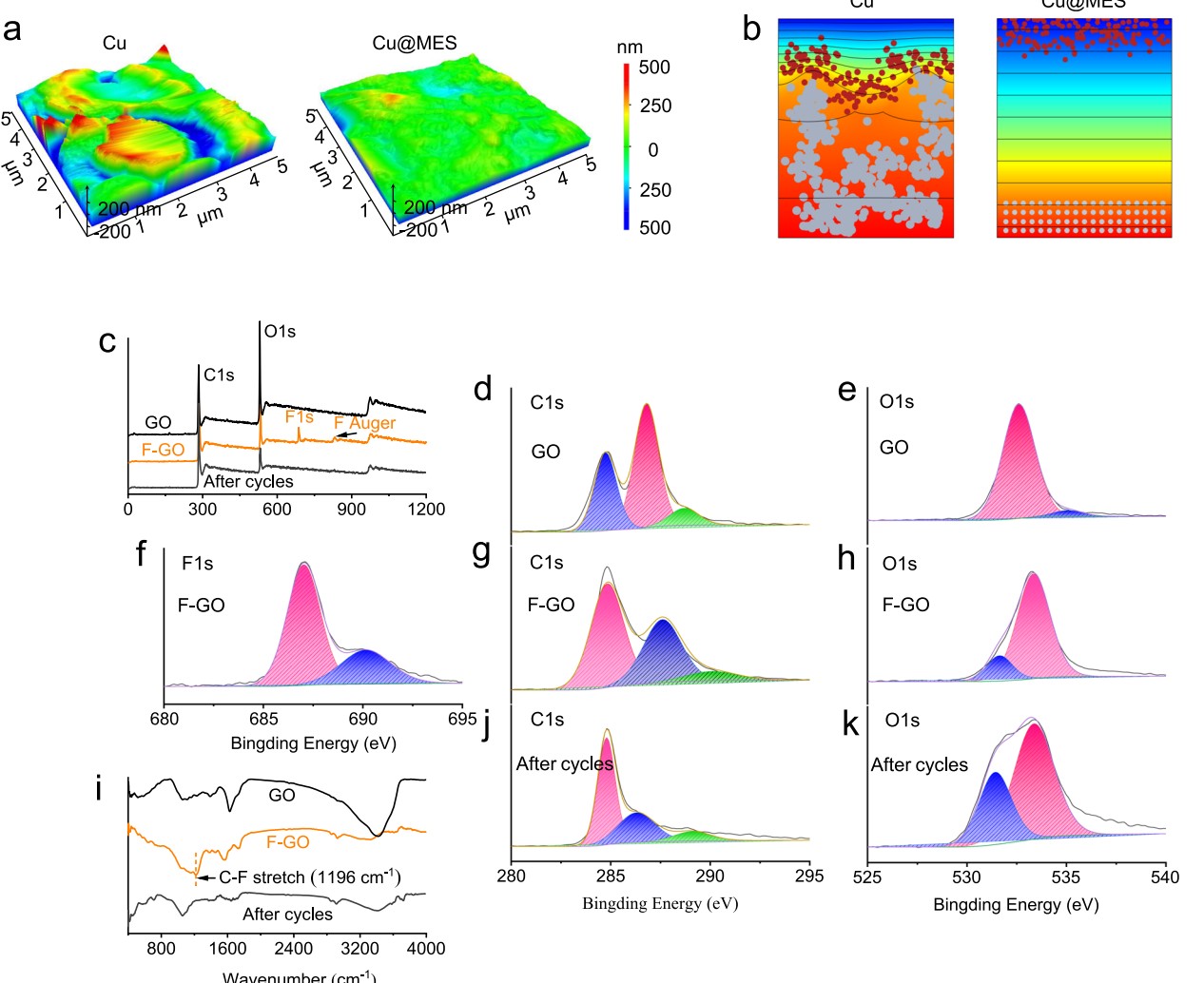

**Fig. 2 | Homogenize electric field and XPS spectra of MES. a** AFM images of Cu foil surface and F-GO. **b** Surface electric field and corresponding ion distribution of the pole piece electrode during initial K plating. **c** XPS survey spectrum. **d**, **e** The high-resolution XPS spectrum of GO. **f–h** The high-resolution XPS spectrum of F-GO. **i** Infrared spectrum of GO, F-GO and F-GO after the cycles. **j**, **k** The high-resolution XPS spectrum of F-GO after the cycles.

concentrated at the tip of the rough K substrate due to the enhanced local current density, known as the "tip effect". Metal anodes with an interface modification can greatly enhance the surface flatness and suppress dendrite formation (Fig. 1b). The interfacial protection strategy is effective to realize dendrite-free PMB. Here, inspired by the structure and function of human skin, a skin-like bionic protective layer MES is constructed on the surface of K metal to protect the metal electrode by improving the interface stability and inhibiting the growth of dendrites (Fig. 1c, d). MES is composed of GO and enhances the in situ formation of SEI (Fig. 1e). The core material of MES is fluorine-doped graphene oxide, and the synthesis method is shown in Supplementary Fig. 1a, b. First, the GO (Supplementary Figs. 2a–h and 3a, b) was fluorinated (Supplementary Figs. 4a–h and 5a, b), and the oxygen-containing functional groups (including carboxyl groups, hydroxyl groups, etc.) in GO were partially replaced by fluorine atoms to generate F-GO (Supplementary Table 1 and Table 2). As shown in Supplementary Fig. 6, F-GO obtained after treatment was dispersed in an alcohol solution and formed a homogeneous suspension after sonication. The suspension was slowly dropped into the deionized water, and the F-GO on the surface was slowly extracted by Cu foil (Supplementary Fig. 7a–d). After multiple extractions, a membrane with a certain thickness was obtained (Supplementary Fig. 8a–d). The repeated folding experiment shows that F-GO possesses a superior

metal fatigue performance (Supplementary Fig. 9a–f). The dry Cu@F-GO and K foil (Supplementary Fig. 10a–d) are rolled together using a roller press. This preparation method is facile and scalable, suitable for the large-scale preparation. PMB covered with F-GO was directly used to assemble coin cell and pouch cell. The deposition of K during cycling can promote the cleavage of the carbon-fluorine bond, forming a fluoride-rich SEI on the surface. The graphene and SEI function simultaneously to form an MES.

**Improve surface flatness and homogenize electric field**
Similar to the dermis and epidermis of human skin which protect internal cells and muscles, the GO layer and SEI layer on the surface of K metal are able protect the metal anode. An important role of the MES on the metal anode is to largely increase the flatness of electrode surface (Supplementary Fig. 12a–d). Using Cu foil as a control sample (the surface flatness of the treated K foil is similar to that of Cu foil), SEM and atomic force microscopy (AFM) were adopted to study the surface structure of the MES and Cu foil. From SEM image in Supplementary Fig. 7 and Supplementary Fig. 10, the surface of the Cu foil is similar to the bare K foil. Through the three-dimension (3D) and two-dimension (2D) AFM images (Fig. 2a, Supplementary Fig. 13a, b), we can visually see the wrinkled microstructure on the surface of the Cu foil, which displays large surface undulations and poor flatness. SEM

images after the material coverage show an increase in surface flatness and uniformity. The treated Cu foils were also investigated using AFM. Both 3D and 2D testing results exhibit high a flatness (Fig. 2a, Supplementary Fig. 13c, d), providing a positive impact on the initial deposition of metal. In addition, the contact angle tests performed under two kinds of electrolyte (3 M KFSI in DME and 0.8 M KPF$_6$ in EC/DMC, EC/DMC = 1:1), indicate that the modified surface can enhance the wettability of the electrolyte (Supplementary Fig. 14a, b). To further illustrate the effect of surface flatness on metal plating, we used Comsol simulations to investigate the effect of surface flatness on the potential distribution and K$^+$ deposition morphology. As shown in Fig. 2b, the electric field intensity changes faster at the tip position. The vicinity of the electrode tip has a tighter potential profile, implying a stronger electric field near the tip. The tip effect causes K$^+$ to be attracted by electrostatic forces and accumulate at the tip. In the case of MES coverage, the uniformity of the surface field strength is greatly enhanced. The corresponding distribution of K$^+$ near the surface is more uniform due to the increase of surface flatness. The simulation model verifies that the surface flatness increases and the surface electric field become more uniform. The uniform interfacial electric field affects the ion distribution and generates a uniform deposition on the electrode surface. Improved surface electric field uniformity benefits the initial plating of K metal, suppresses the growth of dendrites, prolongs the lifespan of metal batteries and enhances the safety.

## In situ formation of enhanced SEI

To explore the in situ SEI enhancement process in MES, computational studies using density functional theory (DFT) were performed to determine whether K deposition could promote the cleavage of carbon-fluorine bonds. K and graphene fluoride unit cells were modeled as the original reactants, while KF and graphene unit cells were modeled as the products. Supplementary Fig. 15 and Supplementary Table 3 show the lattice parameters of each modeled substrate. The calculations display that the carbon-fluorine bond will break automatically with a contact of K and fluorinated graphene, generating KF and graphene. The insertion of each K atom releases an energy of 3.87 eV. In the optimized structure, K promotes the breaking of carbon-fluorine bonds in F-GO, forming a graphene layer. As the fluorine is completely extracted, the hybrid structure of graphene changes from the initial $sp^3$ hybrid structure to $sp^2$, and the overall structure becomes more planar. The carbon-fluorine bond energy in F-GO is lower than that in fluorinated graphene, which is easier to break down. The increase of fluorine amount will form a fluorine-rich SEI on the surface, which enhances the mechanical strength and stability of MES. To further verify the change of fluorine element in F-GO, X-ray photoelectron spectroscopy and infrared spectroscopy were used to characterize the material under different states. As shown in Fig. 2c, F-GO is able to introduce fluorine element compared to the pristine GO (Fig. 2d–h). The infrared spectrum results in Fig. 2i and high-resolution XPS spectrum of cycled F-GO (Fig. 2j, k) also verify the calculation results. Testing on the cycled material demonstrates the disappearance of fluorine, which is consistent with the DFT calculations. The asymmetric cells were disassembled after 10 cycles of charge and discharge, and SEI on the surface of Cu foil was characterized. The composition and content of SEI were obtained by testing the surface of different materials by XPS. As shown in Supplementary Fig. 16a–c, in addition to the conventional sulfides, carbonates, nitrides and oxides, fluorides are clearly distinguished (Supplementary Fig. 17a–c). This also indirectly proves that the fluorine element in F-GO can be transferred into the SEI on the surface, enhancing the mechanical properties and stability of the SEI. To further verify that K plating promotes the formation of enhanced SEI on the electrode surface, XPS spectra tests were performed on surface SEI with different plating capacities. As shown in Supplementary Fig. 18a, in addition to the conventional sulfides,

carbonates, nitrides and oxides, fluorides are clearly distinguished. With the increase of deposition area capacity, the content of F on MES surface increases gradually. This means that the more K is deposited, the more fluorine is released from F-GO. The mechanical strength of SEI can be improved effectively by increasing the inorganic content of SEI (Supplementary Fig. 18b). In addition, the MES surface deposited at 0.5 mAh cm$^{-2}$ was etched at various depths and then XPS was performed (Supplementary Fig. 19a). The content of F at different depths is basically the same, which is higher than that on the GO surface (Supplementary Fig. 19b). This indicates that the enhanced SEI has a uniform F distribution, and the increase of inorganic component benefits the mechanical strength of SEI. TEM and the HRTEM images of the SEI intuitively illustrate the mechanical strength and stability of the SEI on MES. As shown in Supplementary Fig. 20a–f, the thickness of the SEI on the GO surface becomes uneven and cracks appear after many cycles. However, the thickness of SEI on MES surface is uniform. Benefiting from the increase of F content, the enhanced SEI with a high mechanical stability was confirmed by evaluating its modulus with different depth (5, 6 and 7 nm). As shown in Supplementary Fig. 21a, with the increase of indentation depth, the modulus of SEI of F-GO surface increases sharply. At 5, 6 and 7 nm, the modulus of SEI are 5.3, 8.9 and 14.2 GPa, respectively (Supplementary Fig. 21b). Correspondingly, the SEI on the GO surface has only 2.8, 3.6 and 4.5 GPa at the same position (Supplementary Fig. 21c). The results indicate that the increase of fluorine content greatly improves the mechanical strength of SEI, which plays an important role in preventing SEI breakage and inhibiting dendrite generation. The stable and dense SEI protects the electrode from the corrosion of the electrolyte (Supplementary Fig. 22). To demonstrate the protective effect of MES on the electrode, the potential holding test of the Cu@MES electrode at 4 V vs. K$^+$/K was carried out. In the test, Cu@MES||K was able to maintain a stable voltage of 4 V while the current was zero, indicating that Cu is protected by MES and electrolytes do not penetrate F-GO. As a comparison sample, the bare Cu failed to maintain a normal voltage of 4 V during the test, and the current in the battery gradually increased to the instrument's maximum range, suggesting that the bare Cu was seriously corroded. To verify the element content of SEI, element mapping was used. The content of F element in SEI on the surface of GO (Supplementary Fig. 23 and Supplementary Table 4) is lower than that of MES (Supplementary Fig. 24 and Supplementary Table 5), which is in good agreement with the XPS analysis results. The cleavage of the carbon-fluorine bond results in the formation of a fluorine-rich SEI on the K metal surface. The stability of the metal surface is improved and the growth of dendrites is suppressed. The cycled F-GO was treated to eliminate the influence of electrolyte and SEI, and then TEM and EDS mapping tests were performed. Supplementary Fig. 25a–h indicate that the main structure of the material is changed, and the fluorine content drops significantly (Supplementary Fig. 26 and Supplementary Table 6). This agrees with the above calculation and representation structure, which further validates the conjecture.

## Enhanced stability of metal battery by MES

To reveal the reaction kinetics of K anodes, symmetric cell fabricated with MES, GO layers, and bare K electrodes were analyzed by cyclic voltammetry (CV). Figure 3a demonstrates a typical centrosymmetric loop, reflecting the electrode's highly reversible electrochemical behavior. Compared with the GO and bare K electrode, the metal electrode with the MES shows a more pronounced current, demonstrating the reversible plating and stripping of K. In addition, Tafel curves (Fig. 3b) show that the K@MES electrode has a higher exchange current density compared with the K@GO and bare K electrodes, indicating a higher charge transfer rate. Electrochemical impedance spectroscopy (EIS) tests show (Fig. 3c) that the impedance of K@MES is much lower than that of K@GO and bare K among the three

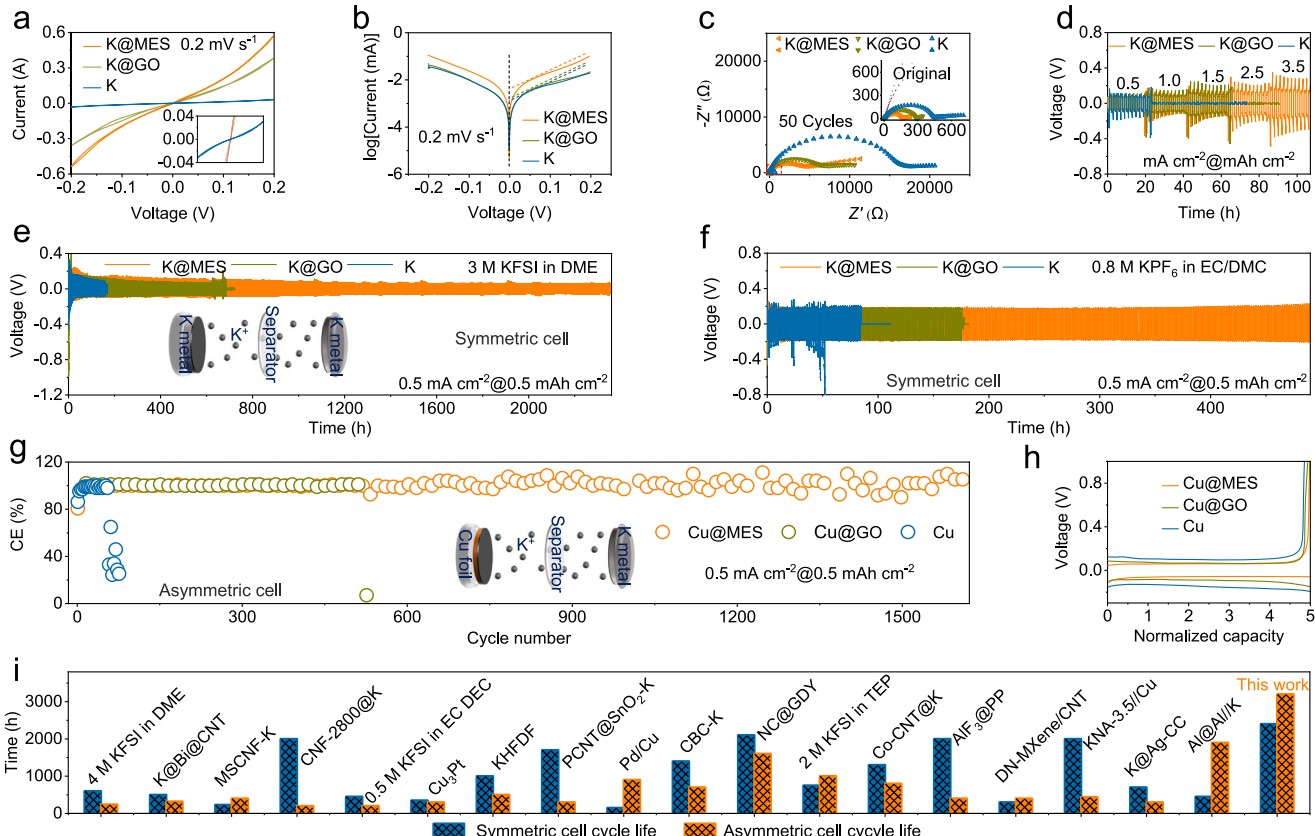

**Fig. 3 | Electrochemical performance of K metal batteries. a** CV curves. **b** Tafel curves and **c** electrochemical impedance spectra curves of symmetrical cells recorded at a scan rate of 0.2 mV s⁻¹ between −0.2 and 0.2 V. **d** Voltage profiles of symmetric cells at various current densities. **e** K plating/stripping voltage profiles for the symmetric cells at a fixed current density of 0.5 mA cm⁻² and capacity of 0.5 mAh cm⁻² with KFSI electrolyte and (**f**) KPF₆ electrolyte. **g** CE as a function of cycle number for Cu||K asymmetric cells with KFSI electrolyte and (**h**) corresponding voltage profiles. **i** Comparison of the Cu||K and K||K cycle life between the K@MES and recently reported PMB.

symmetric cells. After cycling, the battery composed of K@MES electrode maintains the minimum interfacial impedance, which is the result of the evolution of K deposition and SEI during cycling. In addition, symmetrical batteries with different number of cycles were tested. A symmetrical cell composed of K@MES exhibits a stable low impedance, while K@GO and K exhibit larger and unstable impedance. This shows that stable SEI plays an important role in stabilizing the interface (Supplementary Fig. 27). The rate performance of symmetric cells were tested by different current densities and capacities. As shown in Fig. 3d, three symmetric cell was tested at current densities of 0.5, 1, 1.5, 2.5 and 3.5 mA cm⁻² and deposition capacities of 0.5, 1, 1.5, 2.5 and 3.5 mAh cm⁻². The K@MES exhibits excellent interfacial properties and fast K⁺ migration kinetics, as well as excellent dendrite suppression at large current densities and capacities. In contrast, battery with K@GO and bare K anodes unveil larger overpotentials and fluctuations in plating/stripping voltage curves at large current density. Besides, the symmetric cell composed of K@GO was short-circuited under the test conditions of 1.5 mA cm⁻² and 1.5 mAh cm⁻², while symmetric cell composed of bare K was short-circuited under the test conditions with a slightly increased current density and capacity (1 mA cm⁻², 1 mAh cm⁻²). Under the test conditions of a current density of 0.5 mA cm⁻² and a deposition capacity of 0.5 mAh cm⁻² (Fig. 3e), the plating/stripping lifetime of K@MES is about 2300 h (3 M KFSI in DME). However, the batteries with K@GO and bare K anodes showed a poor cycling life. The symmetric cell composed of K@GO failed after cycling for about 670 h, while the symmetric cell with bare K could only cycle normally for 160 h. This is due to the continuous growth of dendrites at the K metal interface with a dramatic increase in interfacial

instability, leading to a rapid short-circuiting of symmetric cell. In the traditional low-concentration electrolyte (0.8 M KPF₆ in EC/DMC, 1:1), the K@MES symmetrical cell is still superior to other two K anodes (Fig. 3f). It is worth noting that different electrolyte systems still show the better stability and longer cycle life of MES, which proves the universality of MES. The CE of the battery was quantified by preparing an asymmetric cell, as shown in Fig. 3g. Cu@MES||K offers a more stable cycling performance compared with Cu@GO||K and Cu||K longer life. Cu@GO||K and Cu||K can only provide 300 and 60 cycles at a current density of 0.5 mA cm⁻², and the CE of the battery drops sharply, which is due to the severe volume expansion and dendrite growth accompanied by the production of dead K. While Cu@MES||K achieves for over 1600 cycles (over 3200 h) under the same test conditions, which is the longest cycle life for this type of battery to date. Benefiting from the dual protection strategy of MES, the cycling stability of the PMB is significantly improved. Furthermore, Fig. 3h presents the plating/stripping curves of three different cells under the same cycling conditions. The nucleation overpotential of the asymmetric cell with Cu@MES is lower than that of Cu@GO and bare Cu, which further demonstrates the excellent ability of MES to stabilize the interface. Such a long cycle life is the best result reported for a PMB (Fig. 3i)[17,21,23,26,27,33–45]. To further verify the universality, MES was fabricated on the surface of lithium metal in the same way and a symmetric cell was assembled. The cycle life of the Li@MES assembled symmetric cell exceeds 4000 h (Supplementary Fig. 28a, b), which is far superior to the symmetric battery composed of Li@GO and bare Li. Even with a commercial conventional low-concentration electrolyte (0.8 M LiPF₆ in EC/DEC/DMC), the symmetric cell composed of Li@MES still

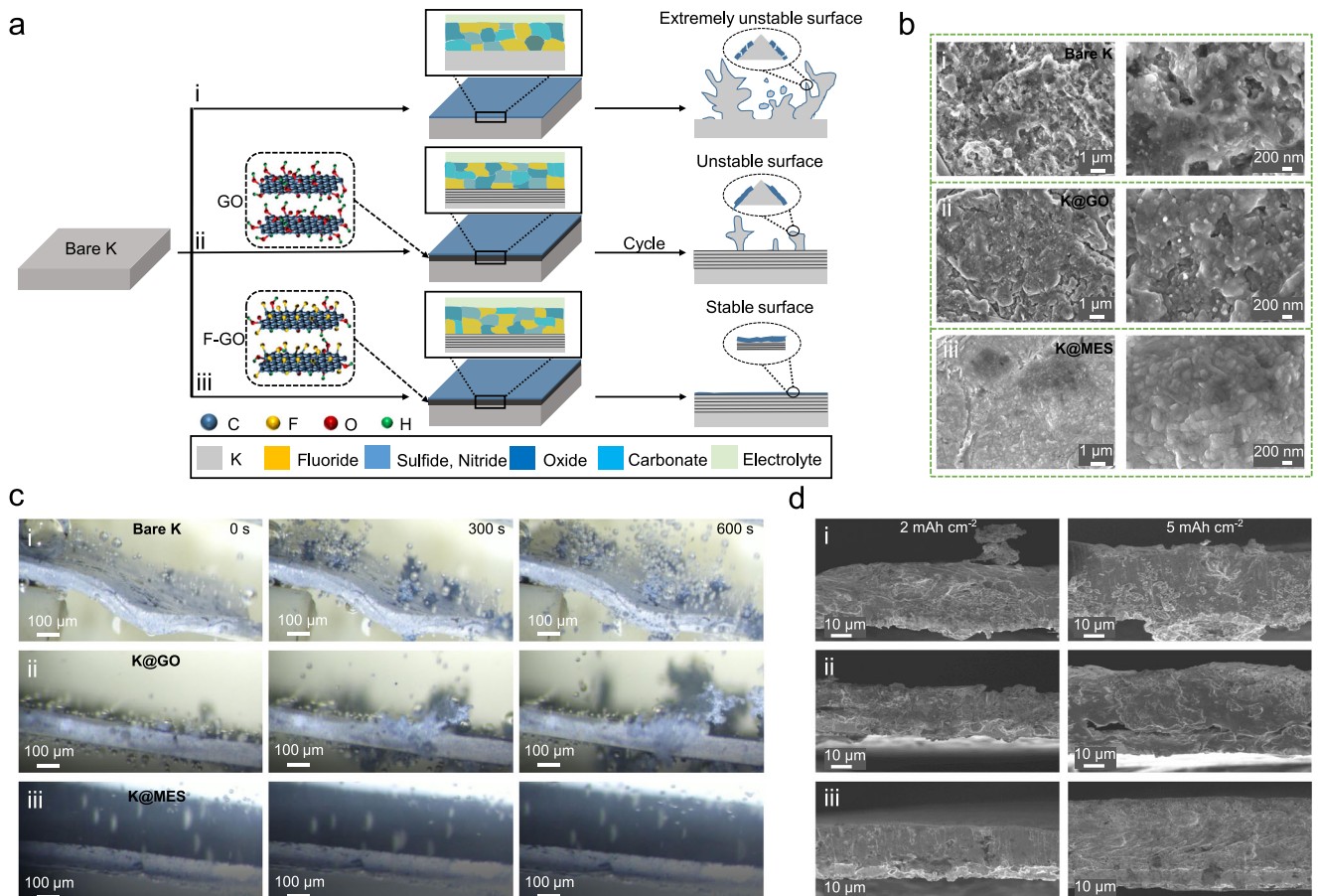

**Fig. 4 | Interfacial evolution and surface characterization of metal electrodes.** **a** Schematic diagram of the interfacial evolution of different forms of K metal. **b** SEM image of different forms of K metal surface after cycling. **c** In situ optical microscopy images of K deposition at different times on the surface of three different PMB. **d** SEM cross-sectional images of K metal deposited on different current collectors (0.5 mA cm⁻²). ((i) corresponds to bare K, (ii) corresponds to K@GO, and (iii) corresponds to K@MES).

present advantages in the cycle life and polarization voltage (Supplementary Fig. 29a, b). The CE of the battery is quantified by preparing asymmetric cell, as shown in Supplementary Fig. 30a, Cu@MES||Li exhibits a more stable cycling performance and longer cycle time compared to Cu@GO||Li and Cu||Li. Meanwhile, the asymmetric cells fabricated by Cu@MES||Li generate lower nucleation overpotentials (Supplementary Fig. 30b). The rate performance of Li@MES symmetric cells was tested by different current densities and capacities. As shown in Supplementary Fig. 30c, the three comparative symmetric cells were tested at current densities of 1, 3, 5 and 10 mA cm⁻² and deposition capacities of 1, 3, 5 and 10 mAh cm⁻². Li@MES exhibits excellent interfacial properties and fast Li⁺ migration kinetics. In addition, MES was applied to sodium metal batteries in the same way. Tested either in NaPF₆ (Supplementary Fig. 31a, b) or NaFSI (Supplementary Fig. 32a, b) electrolyte, symmetrical cell assembled by Na@MES demonstrated the longest cycle life. These tests not only demonstrate the excellent performance of MES in suppressing dendrites and stabilizing the interface, but also reveal the universality of MES in metal anode applications.

## MES stabilizes the interface and inhibits dendrites

To further demonstrate the superiority of MES, a more direct method was used to observe the surface morphology of K metal. Figure 4a schematically depicts the results by changing the interface of PMB in various ways. For bare K, several K components, including KF, form on the surface. But the initially formed SEI is unstable, giving rise to uneven K plating and stripping. After cycles, the unstable interface becomes worse, which exacerbates the dendrite growth (i in Fig. 4a). The use of GO-protected K metal interface can increase the metal surface flatness, which is beneficial to the uniform deposition of ions. GO can temporarily act as a host of metal plating, alleviating the stress changes caused by volume changes and protecting the integrity of the SEI. However, it cannot withstand a long-term cycling, and the naturally formed SEI still cannot effectively protect the interface (ii in Fig. 4a). To enhance the interfacial stability, the use of F-GO for interfacial modification increases the surface flatness of the metal anode, reaching the uniform surface electric field and ion concentration. The deposition of K metal into the interior of F-GO also alleviates the volume expansion of the interface. K can promote the cleavage of carbon-fluorine bonds and release a large amount of fluorine to from a fluorine-rich SEI surface. Two parts of the interface from the MES greatly improve the interfacial stability, suppress the dendrite growth, and significantly boosts the battery cycle life and stability (iii in Fig. 4a). SEM images illustrate the effects of these three treatments on the K metal surface. As shown in Fig. 4b, under the same test conditions, a large number of dendrites appear on the bare K surface. Interfacial instabilities during deposition also result in highly loose and porous surfaces (Supplementary Fig. 33a–d). The non-uniform flux distribution and poor interfacial stability. After surface modification with GO, even though the problems of dendrites and the surface porosity are alleviated (Supplementary Fig. 34a–d), but they are not solved. After using the K metal surface of the MES for K deposition, a smoother surface evolves, and the problem of loose and porous metal electrodes is solved

(Supplementary Fig. 35a–d). It is further demonstrated that the MES plays a significant role in inhibiting dendrite growth and stabilizing the interface. Furthermore, as shown in Fig. 4c, the dynamic deposition process of K can be directly observed using in situ optical microscopy. Other things being equal, the surface morphology changes of three different metal anodes during the deposition process were investigated. Within 300 s of plating, multiple dendrite growth points appear on the bare K surface. A few dendrite points show up on the surface of the K@GO, while the MES-protected metal surface displays no change compared with the initial state. When the plating time extends to 600 s, a large amount of mossy K and dendrites emerge on the bare K surface, accompanied by the production of dead K. Although the dendrite points of the GO-modified metal anode do not increase, the dendrite growth is extremely fast, which may cause fatal damage to the metal battery. In contrast, the metal anode with MES remains dendrite-free. This indicates that K is uniformly and densely deposited on the metal anode under the protection of the MES, which can inhibit the formation of dendrites during a long-term cycling and prolong the cycle life of K metal battery. K@MES was taken out after cycling for SEM test. By observing the surface and cross section of the electrode and characterizing through EDS, MES can be retained on the surface of K metal in the state of stripping (Supplementary Fig. 36a–d) and plating (Supplementary Fig. 37a–d). These results indicate that MES can continuously protect metal electrode, maintain structural stability and surface flatness after several cycles. In addition to above characterizations, K metal with the same capacity was also deposited on the Cu foil, and the ability of MES to suppress dendrites and stabilize the metal electrode was verified by cross-sectional characterization. On the surface of three kinds of Cu foil, $2\,mAh\,cm^{-2}$ of K metal was deposited, as shown in Fig. 4d and Supplementary Fig. 38a–c. The Cu@MES surface presents dense and dendrite-free morphology. On the Cu@GO and bare Cu surfaces, the large volume expansion and dendrite formation occurred during the deposition of K metal. Even if the deposited surface capacity increases to $5\,mAh\,cm^{-2}$ (Supplementary Fig. 39a–c), the Cu foil modified by MES still maintains a clear advantage compared with Cu@GO and bare Cu. As shown in Supplementary Fig. 40a–c, the surface of Cu@MES remained smooth and clean without dendrites after the battery was disassembled, while the inhomogeneity of K metal on the surface of Cu@GO and bare Cu was obvious. These evidences strongly demonstrate the excellent ability of MES to suppress dendrites formation and thereby stabilize the interface. In addition, to further demonstrate the protective effect of MES on K metal, we tested metal electrodes in a more aggressive manner. The recycled K@MES was removed from the PMB and placed in an air environment to test the heat distribution of the metal electrodes using an infrared thermal imager. As shown in Supplementary Fig. 41a, b, at the initial stage, the bare K electrode reacted violently with the air, releasing a large amount of heat (0 min), while the metal electrode protected by MES could isolate the air and slow down the reaction rate. With the increase of reaction time, the bare K electrode maintained the violent reaction with the air, showing a high temperature. K@MES electrode remained stable (30 min) after a long exposure to air, demonstrating that MES can protect K metal and improve the safety of PMB.

### Enhanced electrochemical performance of metal full battery by MES

To verify the application potential of MES, PMB treated with different ways was adopted to match the full battery with PB, and the performance and cycle life were evaluated with KFSI electrolyte. Figure 5a shows the charge/discharge curves for the three batteries after 500 cycles and there is a clear difference in specific capacity. In addition, the full cell using K@MES as the anode also exhibits less polarization voltage due to the advantages brought by the interfacial stability. The

long-cycle performance test results of different full cells at specific current of $500\,mA\,g^{-1}$ are shown in Fig. 5b. The full cell using the K@MES can provide a reversible capacity more than $110\,mAh\,g^{-1}$, while other two PMBs provide slightly lower reversible capacities due to a larger polarization. However, after 500 cycles, the stability of the K metal interface modified by graphene oxide decreases, and the CE drops. The full cell composed of bare K experiences a faster capacity decay, with the capacity decaying to $58\,mAh\,g^{-1}$ after 500 cycles. Using $KPF_6$ electrolyte, K@MES is able to maintain a stable cycling over 500 cycles under the same conditions. Due to the interfacial instability, CE of the full cell using K@GO and bare K decays significantly after 150 and 100 cycles (Supplementary Fig. 42a, b). As shown in Fig. 5c, the full cell was tested for the long-cycle performance at a specific current of $1000\,mA\,g^{-1}$ with a cycle life of over 5000 cycles. During cycling, CE of the full-cell exceeds 98%. By comparing with recently published literature (PMBs with modified anode interface or PB for cathode), K@MES shows a great advantage in the cycle life. The long cycle life of the full cell is an important step towards the commercialization of PMBs (Fig. 5d)[17,21,23,25,27,38,40,41,46–48]. Therefore, the full battery assembled using K@MES modification exhibits a stable long-cycle stability, which makes K-ion batteries promising for commercial applications. In addition, in the rate performance test, the full cell with K@MES provided reversible capacities around 110, 93, 81 and $75\,mAh\,g^{-1}$ at specific current of 500, 1000, 2000 and $3000\,mA\,g^{-1}$, respectively, while other two full cells presented lower capacities under the same test conditions (Fig. 5e). When the electrolyte was changed into $KPF_6$, the rate performance and charge-discharge curve comparison of the full cell is basically the same with KFSI (Supplementary Fig. 42c, d). In addition, we also assembled pouch cell for performance testing (Fig. 5f). To understand the stable cycling of the pouch cell, the corresponding voltage profile was observed (Fig. 5g). All of them show a stable voltage curve with the charge/discharge intersection in the curve slightly shifted to the left horizontally, which is a good indication of the stable electrochemical performance of the K@MES anode in the pouch PMB. The pouch cell, composed of a K@MES anode and PB cathode, was stable for 100 cycles at a specific current of $1000\,mA\,g^{-1}$ (Fig. 5h). Above results indicate that MES-modified metal electrode has a better electrochemical performance than the GO-modified and bare K, which is attributed to the protection of the MES. Overall speaking, this study consistently demonstrates that K@MES exhibits an excellent electrochemical performance.

## Discussion

In conclusion, we successfully designed a skin-like biomimetic protection layers for metal electrode through a simple and effective strategy. The artificially and in situ constructed skin-like dual protective structure can inhibit dendrite growth on K metal at the high deposition capacity and high current density to prolong the cycle life of metal batteries. The modified PMB exhibits a significant enhancement of electrochemical performance both in symmetric cell and full cell. K symmetric cell reached a stable cycling for 2300 h. In addition, the asymmetric cell composed of MES-modified Cu foil can be stably deposited and stripped for over 3200 h under a current density of $0.5\,mA\,cm^{-2}$ and a capacity of $0.5\,mAh\,cm^{-2}$, achieving an extraordinary cycle life. Meanwhile, the corresponding full cell has the superior rate capability (a capacity of $75\,mAh\,g^{-1}$ at $3000\,mA\,g^{-1}$) and cycle life (over 5000 stable cycles at $1000\,mA\,g^{-1}$). This strategy provides a novel style for the design and fabrication of novel biomimetic metal electrode interface, facilitating the multidisciplinary collaborative research.

## Methods

### Materials synthesis

Diethylaminosulfur trifluoride (DAST), o-dichlorobenzene (DCB), methanol, ethanol and acetone were purchased from Sinopharm Chemical Reagent Co. Ltd. (SCRC). The K metal comes from songjing

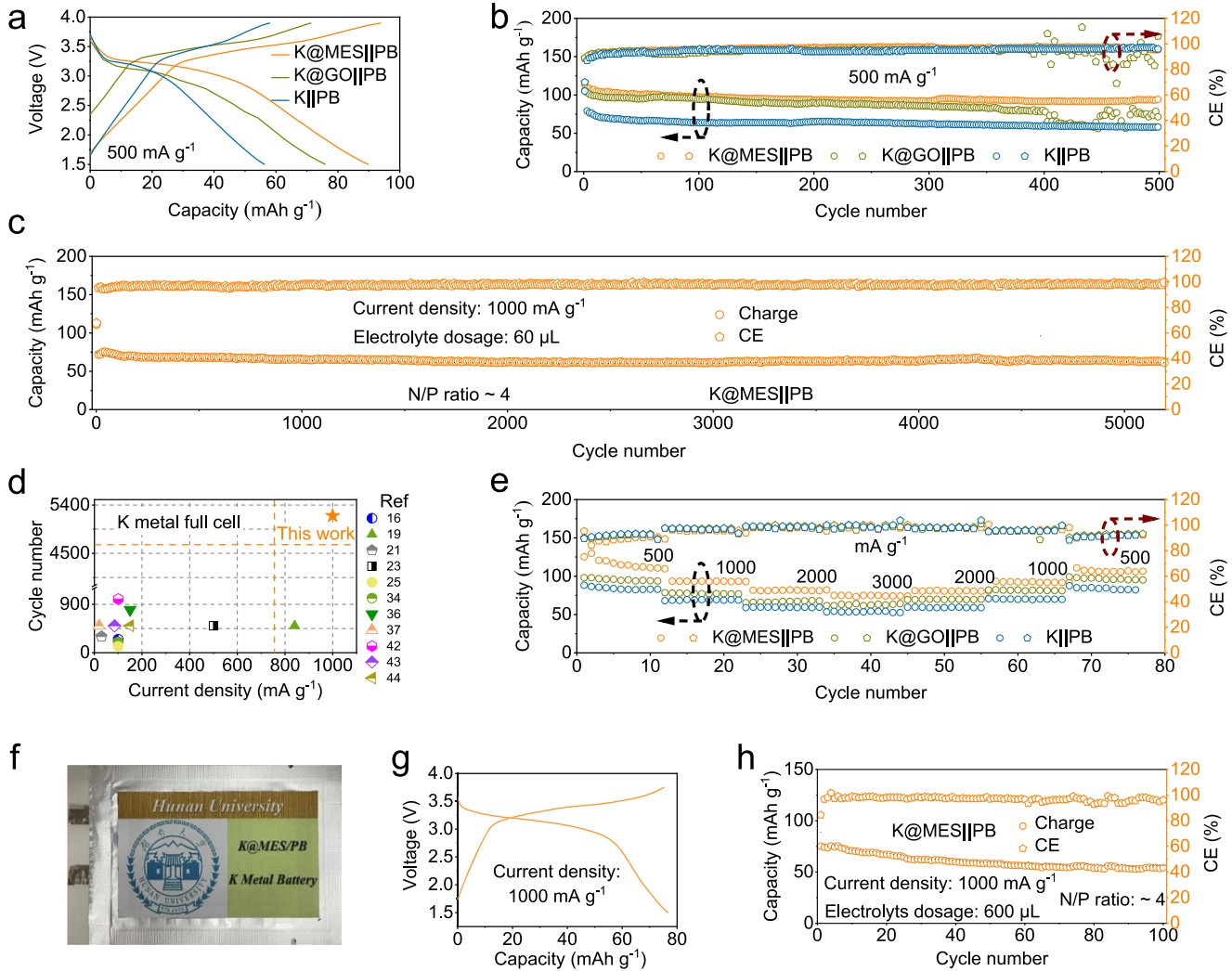

**Fig. 5 | Electrochemical performance of K∥PB coin cells with KFSI electrolytes.** **a** Charge and discharge voltage profiles of the K∥PB cells and (**b**) cycling performances at a specific current of 500 mA g⁻¹. **c** Long-term cycling performance of K∥PB cells coin cells with K@MES anodes at a specific current of 1000 mA g⁻¹. **d** Recently published cycle life and current density comparisons for K metal full cells. **e** Rate performance of the K∥PB cells. **f** Pouch cell consisting of K@MES. **g** Corresponding voltage profiles of the K@MES∥PB on 1000 mA g⁻¹. **h** Cycling profiles of the discharge specific capacity (left y axis) and the CE (right y axis) for the K@MES∥PB on 1000 mA g⁻¹ with pouch cell.

New Energy Co. LTD. Graphene oxide (GO) was purchased from Suzhou Carbon Fung Graphene Co., Ltd. Deionized water (DIW) was used in all experiments. All of the chemicals used in this experiment were analytical grade and used without further purification.

## Preparation of F-GO

Fluorine doping graphene oxide (F-GO) was prepared by a mild liquid-phase reaction between GO and DAST. In a typical procedure, DCB (150 mL) and GO powder (0.15 g) were added sequentially to a 250 mL PTFE flask. The mixture was vigorously stirred overnight to facilitate GO dispersion. After that, DAST (2.5 mL) was added dropwise over 10 min at 0 °C. The reaction mixture was first sonicated for 6 h and then stirred at ambient conditions for 3 days. Finally, the reaction was quenched by the careful addition of 100 mL of methanol. The fluorinated product was obtained by filtering the mixture, followed by thorough washing with ethanol, acetone, and deionized water to remove impurities, and vacuum drying at 60 °C for 12 h.

## Preparation of Cu@F-GO

The F-GO suspension was prepared by dispersing the fluorinated reduced graphene oxide prepared above in pure ethanol, followed by

sonication for 60 min. The desired metal electrode skin host material was constructed on Cu foil by the Langmuir–Blodgett method. Briefly, Cu foils were cut to any size. Then the foils were immersed in a beaker filled with water, and the desired F-GO suspension was continuously injected on the water surface to fabricate the metal electrode skin host material. When the self-assembled film covered about 70% of the water surface, by simultaneously lifting the Cu foil immersed in the water. The material with the desired thickness was finally obtained by repeating the above process. The Cu@ F-GO was then finally dried on a hot plate at 80 °C for 3 min and in a vacuum oven at 80 °C overnight. The preparation of Cu@GO was the same as that of Cu@F-GO.

## Preparation of K@F-GO

The bare K metal electrode was prepared first. The raw lump metal K was placed between two plastic films, and then the K and plastic film are placed together in the middle of the roller of the roller press. After adjusting the distance between rollers, the speed of the roller press is set as 0.1 cm s⁻¹, and the uniform K foil was obtained by repeatedly rolling three times. The K foil was used as the control sample in the experiment. The metal electrode skin main material was transferred to

K foil of desired thickness by a rolling process. Cu@F-GO and K foil were stacked together and sandwiched between two plastic films. The stacked components were then uniformly pressed using a roll press. The roll distance is about 80% of the total thickness of the stacked components, and the rolling speed is 0.1 cm s$^{-1}$. After the roll press, the plastic film was removed and the Cu foil was peeled off to obtain K@F-GO. The preparation of K@GO was the same as that of K@F-GO. Bare K metal electrodes, K@GO and K@F-GO were cut into discs with 1.2 cm diameter (area is 1.13 cm$^2$) for coin cell (CR2032) measurements. All of the above steps were done in a glove box environment (water and oxygen content are less than 0.5 ppm).

### Material characterization

The morphology of the material was investigated by transmission electron microscopy (TEM, Titan G2 60-300 with image corrector). Field emission scanning electron microscope (FESEM, Hitachi S-4800, 20 kV) images revealed the shape of the material and dendritic, and the elemental content of materials were revealed through energy dispersive spectroscopy (EDS). The in situ optical microscopy was used to further observe the growth process of K dendrites. The heat release of K metal in air was tested using a thermal imager.

### Electrochemical measurements

The symmetric cell consisting of two identical bare K metal electrode, K@GO and K@F-GO electrodes were assembled in coin cell (CR2032) and tested at current density of 0.5 mA cm$^{-2}$ with the plating-stripping capacity of 0.5 mAh cm$^{-2}$. Asymmetric cell consisting of bare Cu, Cu@GO and Cu@F-GO electrodes and bare K were assembled in coin cells (CR2032) to test the deposition stripping capability and Coulomb efficiency of the asymmetric cells at current density of 0.5 mA cm$^{-2}$ with the plating-stripping capacity of 0.5 mAh cm$^{-2}$. The electrochemical performance of full cell was evaluated by preparing coin cell. The active PB particles were prepared according to the previous literature report with some modification, conducting agent (acetylene black) and PVDF binder were mixed (weight ratio of 6:3:1) in N-methyl-2-pyrolidone (NMP) (AR, 99%, Aladdin) solution and stirred overnight. The cathodes were prepared by spreading the slurry onto an aluminum foil current collector, which was then dried at 80 °C for 24 h. PB electrode were cut into 1.2 cm diameter discs for coin cell (CR2032). The active material loading was about ~ 1.5 mg cm$^{-2}$. The thickness of the K metal electrode in the coin cell is about 200 μm. The glass fiber (diameter: 19 mm, thickness: 260 μm, Whatman, GF/D) papers were used as separators, and the 2032-type coin cells were assembled and disassembled in an argon-filled glove box. The preparation of pouch cells was similar to that of a coin cell. The cathode material was coated on the surface of the Al foil, dried and cut to a size of approximately 3 cm * 4 cm. The uncoated side was coated with the battery tab and then packed into the Al foil together with glass fiber of the same size. K@MES was prepared into an electrode of the same size as the cathode, with nickel tab on the back of the K metal. The assembled pouch cells were filled with approximately 600 μL of electrolyte and vacuum sealed using a vacuum sealer. Short-cycle performance and rate performance tests were carried out to calculate the specific capacity according to the mass of the cathode active material. To verify the practicality of the full cell, long cycle tests and pouch cells were calculated on the basis of the total mass of cathode and anode, with strict control of the electrolyte dosage and negative/positive ratio (N/P ~ 4). The potassium bis(fluorosulfonyl)amide (KFSI, purity 98%, 3 mol L$^{-1}$) in an 1,2-Dimethoxyethane (DME), potassium hexafluorophosphate (KPF$_6$, purity 99%, 0.8 mol L$^{-1}$) in ethylene carbonate (EC):dimethyl carbonate (DMC) (1:1, v/v) were used as the electrolyte. The battery performance and galvanostatic charge/discharge of the batteries were performed using the Neware battery test system (BTSCT-3008-TC 5.X. Shenzhen. China). CV curve, Tafel plots and electrochemical impedance spectroscopy (EIS) of coin cells were characterized by using the CHI660e electrochemical workstation. The frequency range of the EIS was from 10$^{-2}$ to 10$^5$ Hz. CV curve and Tafel plots from linear sweep voltammetry tested in the range of −0.2 V to 0.2 V at 2 mV s$^{-1}$.

The preparation method of the lithium metal symmetrical cell was the same as that of the K metal symmetrical cell, except that the separator is replaced by polypropylene, and the electrolyte was replaced either by Lithium Bis(fluorosulfonyl)imide (LiFSI, purity 99%, 3 mol L$^{-1}$) in 1,2-Dimethoxyethane (DME) or LiPF$_6$ (LiPF$_6$, purity 99%, 0.8 mol L$^{-1}$) in ethylene carbonate (EC):dimethyl carbonate (DMC):diethyl carbonate (DEC) (0.25:1:1, v/v/v). The preparation method of the sodium metal symmetrical cell was the same as that of the K metal symmetrical cell, except that the separator was replaced by polypropylene, and the electrolyte was replaced by Sodium Bis(fluorosulfonyl)imide (NaFSI, purity 99%, 3 mol L$^{-1}$) in an 1,2-Dimethoxyethane (DME) or NaPF6 (NaPF6, purity 99%, 0.8 mol L$^{-1}$) in ethylene carbonate (EC): dimethyl carbonate (DMC):diethyl carbonate (DEC) (0.25:1:1, v/v/v). All electrochemical performance tests were carried out in an incubator at 28 °C.

### Electric field homogenization simulation in 3D

Electric currents module of Comsol Multiphysics was employed for electric field homogenization simulation. The resulting electric field distribution was given in Fig. 2b. The simulation of K deposition was performed as follows. Before ion deposition, all K$^+$ are placed on top. Then a bias potential along the y direction is turned on to allow cations to migrate towards the anode. Once the K$^+$ cation contacts with the current collector surface or the previously deposited metal K, it becomes a metal block and remains in a fixed simulation for the remaining time. After the formation of K metal, new K$^+$ was released from random (x) positions at the top to keep the number of K$^+$ constant. When the K metal reaches a preset height, which was a constant in all simulations, the simulation stops.

## Data availability

The all data generated in this study are provided in the Supplementary Information/Source Data file, or from the corresponding authors upon reasonable request. Source data are provided with this paper.

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

## Acknowledgements

B.L. acknowledges financial support from National Nature Science Foundation of China No. U20A20247 and 51922038. H.D. acknowledges financial support from Postgraduate Scientific Research Innovation Project of Hunan Province No. CX20220382.

## Author contributions

B.L. and C.W. conceived the concept and directed the research. H.D., J.W. planned, performed the experiments, and wrote the manuscript. J.Z. gave advice to the research. All authors discussed the results and commented on the manuscript.

## Competing interests

The authors declare no competing interests.
