## [Peer Review File · Nature Communications]

Building electrode skins for ultra-stable potassium metal batteriesREVIEWER COMMENTS

Reviewer #1 (Remarks to the Author):

This manuscript presents a study of fluorinated graphene oxide protective layer for K metal electrodes as negative electrodes for potassium metal batteries. This study provides interesting data; however, the impact is lost by vague explanations of the results and short discussions of the findings. In addition, the advantages of this approach over similar methods proposed in previous studies have not been fully explained. In my opinion, a substantial revision is needed to make this manuscript suitable for publication in Nature Communication. The authors should clarify the points listed below.

1. In Figure 1d, the authors illustrate the formation of SEI on top of the MES and the plating/stripping of K metals under the MES, describing the similarity to human skin. However, this scheme has not been adequately proven by the experimental data. A series of verifications are needed, as shown below.

- The formation of SEI on top of F-GO means that electrolytes do not penetrate F-GO due to desolvation on the SEI, and K^+ ions diffuse through the F-GO. This point should be verified by experiments such as a potential holding test of the Cu@MES electrode at 4 V vs. K^+/K . If the electrolytes do not penetrate F-GO, Cu@MES should not corrode. Please provide experimental data.
- Verify that the MES is retained at the top of deposited K metal in the SEM image in Figure 4d.

2. The ref. 23 also reported that a similar artificially generated protective layer significantly improves the lifetime of K-metal electrodes. Please explain the superiority of this method over such existing methods. The authors should also introduce ref. 23 in more detail in the manuscript.

3. The authors have shown its applicability to Li metal; would MES work for Na metal?

4. There is a notation $K@MES|Cu$ (around page 11), but I think $Cu@MES||K$ may be correct. Please check it.

Reviewer #2 (Remarks to the Author):

In this manuscript, the authors develop a strategy for stabilizing the metal interface with a metal electrode skin (MES) mimicking human skin, which is novel and creative. The combined effect of the ex-situ construction of a graphene interface layer and the in-situ formation of an enhanced SEI to form a metal electrode skin to protect the metal electrode. The MES-protected metal electrode is capable of achieving long cycle lives for both symmetric and asymmetric cells. $K|Cu@MES$ achieves over 1600 cycles, which is one of the best potassium metal asymmetric cells reported to date. The full cell also shows excellent performance and cycling stability compared to previously published papers. Overall, the bionic interface protection approach reported in this manuscript is innovative. The 'biomimetic' concept used in this work provides an important advance in the design of future metal anodes. Therefore, this paper can be accepted by Nature Commun. Some suggestions are as follows.

Comment 1: According to the information provided in the manuscript, the main material of the MES is F-GO. The surface of the F-GO modified metal electrode or copper foil is smooth and has a high degree of flatness, does this affect the electrolyte infiltration?

Comment 2: In Figure 2, the authors used COMSOL simulations to investigate the effect of surface flatness on the potential distribution and K deposition morphology. In the AFM tests and simulations, the authors used Cu and Cu@MES as a comparison, how does this differ from K and K@MES?

Comment 3: Can MES on metal electrodes or Cu foil surfaces maintain structural stability and surface flatness after several cycles?

Comment 4: In the Figure 3c, both $K@MES$ and $K@Cu$ have artificial interfacial protection layers. Why do $K@MES$ and $K@Cu$ assembled symmetrical cells have a lower impedance? How does the impedance of the different types of symmetrical cells change during cycling?

Comment 5: Supplementary Fig. 13a and b shows the lattice parameters of C_{Fx} (x = 1) modelled substrate. Please carefully explain in the manuscript how C_{Fx} (x = 1) differs from F-GO in the manuscript. The authors need to give a rationale or reference in the text for using C_{Fx} (x = 1) for their calculations.

Reviewer #3 (Remarks to the Author):

This manuscript provided a metal electrode skin (MES) method for potassium metal battery (PMB) anodes. The perfectly designed MES can adjust the electric field and promote the in-situ formation of metal-fluoride nanocrystal-rich SEI layers. Interestingly, the ultra-long cycle life was achieved by Cu@MES|K asymmetric cells and K@MES|PB cells. I would like to recommend the publication of this manuscript after addressing the following issues.

1. What's the difference between MES and artificial SEI? Is MES just a new name for artificial SEI?
2. As the author says, 'The skin is divided into two layers, the epidermis and the dermis.' 'The epidermis mainly displays a protection function and the dermis is to make the skin more extensible and elastic.' The protection function of MES has been greatly reflected in the article, and I wonder whether MES is extensible and elastic. I suggest the author do some experiments about mechanical strength and metal fatigue to prove it.
3. Potassium is a particularly active metal. PMB should have a great safety hazard in the application, so what is its application prospect?
4. Based on question 3, I wonder whether the MES method can improve the safety of PMB, and suggest the authors do more supplement-relevant experiments.
5. In the section named 'In situ formation of enhanced SEI', It seems that the available evidence is insufficient for the in-situ variation of enhanced SEI. I suggest that the authors add some in-situ characterization to further illustrate the changes of SEI in charge-discharge cycles.
6. Some related refs should be cited, including Nano-Micro Lett. 2022, 14, 210 and Nature Communications, 2022, 13, 3209.

Point-by-Point Responses to the Reviewers' Comments

Dear Editor and Reviewers.

Thanks for your letter and the reviewers' comments concerning our manuscript entitled "Building electrode skins for ultra-stable metal batteries". Those comments are valuable and very helpful for revising and improving our work. We have studied the comments carefully and have made correction which we hope to meet with approval. For ease of reference, our responses are shown in red, and the resulting changes to the manuscript are in blue. In the following paragraphs, you will find our detailed response to the reviewer' comments point-by-point. The main corrections in the paper and the response to the reviewer's comments are as following:

Reviewer #1 (Remarks to the Author):

This manuscript presents a study of fluorinated graphene oxide protective layer for K metal electrodes as negative electrodes for potassium metal batteries. This study provides interesting data; however, the impact is lost by vague explanations of the results and short discussions of the findings. In addition, the advantages of this approach over similar methods proposed in previous studies have not been fully explained. In my opinion, a substantial revision is needed to make this manuscript suitable for publication in Nature Communication. The authors should clarify the points listed below.

Response: We thank the reviewer very much for the precious comments on this article, which are important to improve this work. We have made detailed modifications based on these specific opinions and suggestions to improve the quality of our

manuscripts. In general, we want to modify the manuscripts to meet your requirements. The specific reply is as follows:

1. • The formation of SEI on top of F-GO means that electrolytes do not penetrate F-GO due to desolvation on the SEI, and K^+ ions diffuse through the F-GO. This point should be verified by experiments such as a potential holding test of the Cu@MES electrode at 4 V vs. K^+/K . If the electrolytes do not penetrate F-GO, Cu@MES should not corrode. Please provide experimental data.

Response: We appreciate this concern raised by the reviewer. As the reviewer mentioned, the formation of SEI on top of F-GO means that electrolytes do not penetrate F-GO due to desolvation on the SEI, and K^+ ions diffuse through the F-GO. According to the reviewer's suggestion, we carried out a potential holding test of the Cu@MES electrode at 4 V vs. K^+/K . As shown in Figure R1, a comparative experiment was conducted between Cu@MES and bare Cu. In the process of test, Cu@MES||K can keep 4 V, and current is zero, indicating that Cu is protected MES and electrolytes do not penetrate F-GO due to desolvation on the SEI, and K^+ ions diffuse through the F-GO. As a comparison sample, the bare Cu failed to maintain a normal voltage of 4 V during the test, and the current in the battery gradually increased to the instrument's maximum range. The test results in Figure R1 are illustrated that electrolytes do not penetrate F-GO due to desolvation on the SEI, and K^+ ions diffuse through the F-GO. The electrolytes do not penetrate F-GO, Cu@MES was not corroded.

Figure R1. Constant voltage test. Potential holding test of the Cu@MES and Cu electrode at 4 V vs. K^+/K (0.8M KPF_6).

Revision: “The stable and dense SEI protects the electrode sheet from the corrosion by the electrolyte (**Supplementary Fig. 22**). To demonstrate the protective effect of MES on the electrode, the potential holding test of the Cu@MES electrode at 4 V vs. K^+/K was carried out. In the process of test, Cu@MES||K can stay at 4 V, and current is zero, indicating that Cu is protected by MES and electrolytes do not penetrate F-GO due to desolvation on the SEI, and K^+ ions diffuse through the F-GO. As a comparison sample, the bare Cu failed to maintain a normal voltage of 4 V during the test, and the current in the battery gradually increased to the instrument's maximum range. It indicates that the bare copper is seriously corroded.”

- Verify that the MES is retained at the top of deposited K metal in the SEM image in Figure 4d.

Response: We thank the reviewer for reminding us this important point. As the interfacial protection material for metal electrodes, MES and should be able to retain

at the top of the deposited metal. We tested the metal electrode again based on the comments from the reviewer and determined that MES retained at the top of the metal during both metal stripping and plating.

Figure R2. Morphology and structure after K@MES stripping 0.5 mAh cm^{-2} . (a, b) SEM image and EDS mapping of surface. (c, d) SEM image and EDS mapping of cross profile.

Figure R3. Morphology and structure after K@MES plating 0.5mAh cm^{-2} . (a, b) SEM image and EDS mapping of surface. (c, d) SEM image and EDS mapping of cross profile.

Figure R2 shows the electrode state of K@MES after stripping in a symmetrical cell (0.5mAh cm^{-2}). Figure R2a provides the SEM image of the electrode surface after potassium metal stripping. It was observed that MES retained at the surface of the metal electrode after the metal was stripped (graphene was clearly observed). To further prove that MES retained at the top of the metal instead of being covered by potassium metal, elemental analysis of the metal electrode surface was performed by EDS. As shown in Figure R2b, element mapping provides a clear and intuitive view of graphene (C) on the surface. In addition, fluorine and sulfur elements contained in the enhanced SEI can also be detected. Figure R2c provides sectional SEM images under the condition of K@MES stripping, which more intuitively proves that MES is

retained at the surface of the metal electrode. The EDS mapping also indicates that MES retained at the top of metal electrode (Figure R2d). Figure R3a, b, c and d show the SEM images of K@MES after plating (0.5mAh cm^{-2}), and the results obtained are the same as those in Figure R2. These results indicate that MES can be retained at the top of potassium metal either in stripping or plating.

Revision: “In addition, the K@MES was taken out after circulation for SEM test. By observing the surface and cross section of the electrode and characterizing through EDS, it can be found that MES can be retained on the surface of K metal in the state of stripping (**Supplementary Fig. 36a-d**) and plating (**Supplementary Fig. 37a-d**). These results indicate that MES can continuously protect metal electrodes.”

2. The ref. 23 also reported that a similar artificially generated protective layer significantly improves the lifetime of K-metal electrodes. Please explain the superiority of this method over such existing methods. The authors should also introduce ref. 23 in more detail in the manuscript.

Response: We are very grateful for the reviewer's suggestions. Based on the reviewer suggestions, we have again researched and explored ref. 23. The ref. 23 is a study of “Tribo-electrochemistry induced artificial solid electrolyte interface by self-catalysis”. In this study, authors first used a self-catalyzed tribo-electrochemistry reaction to construct a continuous and compact protective ASEI on potassium anode. Polytetrafluoroethylene (PTFE), as a strong electronegative material in the triboelectric series, was selected for the tribo electrochemistry reaction with K metal.

The unique SEI construction strategy can be finished in seconds and greatly solve the interface issues between “traditional” ASEI and electrode surfaces with no complicated physical and chemical methods. It is an innovative and creative study. By comparing the two jobs, we find that ours has the following advantages:

1. The safety and effect of the potassium metal electrode after the interface modification is better. The first is the safety in the preparation process. In ref. 23, researchers mentioned “As we all know, PTFE has unusual chemical resistance and only reacts with molten alkali metals and gaseous fluorine at high temperature and high pressure”. However, as shown in Supplementary Movie 1 and Fig. 1a of ref. 23, an explosive reaction happens accompanied by the liquefaction of K and splashing of products when only pressure is applied. Based on this phenomenon, continuous and compact protected potassium (CCPP) anodes were prepared. Potassium metal is a particularly active alkaline metal. In the process of preparing metal electrodes, an explosive reaction happens accompanied by the liquefaction of K and splashing of products. Safety is an important issue that has to be considered in the preparation process. In our present work, except for the last step of transferring the material to the surface of the potassium foil, the rest can be operated in an external environment with a higher safety. In addition, the growth of dendrites is mainly due to the inhomogeneous electric field at the interface. CCPP uses molten potassium metal in the construction process to improve the flatness of the interface and provide a more stable SEI. However, according to the data provided in ref. 23, we can observe that the metal electrode surface is not highly flat, which can lead to inhomogeneous

Editorial Note: Figures R4 and R6 below reproduced from Qin, C., Wang, D., Liu, Y. et al. Tribo-electrochemistry induced artificial solid electrolyte interface by self-catalysis. *Nat Commun* **12**, 7184 (2021). <https://doi.org/10.1038/s41467-021-27494-z>

electric and ionic fields at the interface, resulting in the growth of dendrites with increased probability of dendrite generation (Figure R4 and R5). In our present work, we artificially construct a highly flat interface and form a more stable SEI in situ during cycling. A higher level of flatness can provide a more uniform electric field to suppress the growth of dendrites.

Figure R4. (a, b) Top views of CCPP by SEM. (c) Cross-sectional view of CCPP, the thickness of the SEI is $\sim 10 \mu\text{m}$. (ref. 23)

Figure R5. SEM image of the K@F-GO and bare K. (a-c) Protected and unprotected potassium metal surfaces. (d) Potassium metal cross section with protective layer.

2. MES has advantages in scalability and scale preparation. In ref. 23, as shown in Supplementary Movie 1 and Fig. 1a, an explosive reaction happens accompanied by the liquefaction of K and splashing of products when only pressure is applied. Based on this phenomenon, continuous and compact protected potassium (CCPP) anodes

were prepared. Among the three alkali metals, lithium, sodium and potassium, potassium has the lowest melting point. By applying pressure to potassium, the metal on the surface liquefies and reacts with PTFE to produce stable ASEI. Lithium and sodium metals have high melting points, and it is uncertain whether this approach can be extended to the surfaces of other metal electrodes. In our research work, it is only necessary to transfer the F-GO film prepared in an external environment to the surface of the metal by a roller press, which does not involve a change of metal state (solid-liquid-solid) and can work on the surface of a wide range of alkali metals. In addition, CCPP can be prepared on a large scale under laboratory conditions only, and the K@MES preparation method is relatively simple. This has some advantages over the metal electrode preparation method in ref. 23 for large-scale applications (Figure R6).

Figure R6. Explosive reaction happens accompanied by the liquefaction of K and splashing of products when only pressure is applied. Based on this phenomenon, continuous and compact protected potassium (CCPP) anodes were prepared. (Ref. 23)

3. Advantages of symmetrical cell and full batteries in cycle life. Whether building CCPP or K@MES, the ultimate goal is to suppress dendrite generation and improve cell safety and cycle life. In the long cycle test, the CCPP anodes can keep stable polarization for more than 1000 h. After a stable cycling, the overpotential of the symmetric cell is about 92 mV. Under the same test conditions, the cycle life of the symmetric cell composed of K@MES is up to 2400 h, and the overpotential after stable cycling is about 60 mV. The longer cycle life and lower overpotential prove that MES has stronger protection ability for metal electrodes. In addition, the full cell assembled with K@MES has a longer cycle life compared to CCPP. In the Supplementary Information of ref. 23, the full cell composed of CCPP and PB was able to cycle stably for about 150 cycles. In our research work, the full cell composed of K@MES and PB can be stably cycled for over 500 cycles under the same test conditions, which far exceeds the cycle life of the full cell in ref. 23. The long cycle life of the symmetric cell and full cell also greatly enhances the practicability of K@MES (Figure R7).

	CCPP	K@MES
Symmetric cell (3M KFSI)	1000 h	2400 h
Overpotential	92 mV	60 mV
Symmetric cell (0.8M KPF ₆)	250 h	500 h
Full battery (PB)	150 cycles	500 cycles
Current density	100 mA g ⁻¹	500 mA g ⁻¹
Capacity	75 mAh g ⁻¹	100 mAh g ⁻¹

Figure R7. Comparison of electrochemical properties of CCPP and K@MES.

The ref. 23 developed a self-catalyzed tribo-electrochemistry strategy to construct a

continuous and compact protective layer on the K electrode surface. The unique SEI construction strategy can be finished in seconds and greatly solve the interface issues between traditional ASEI and electrode surfaces with no complicated physical and chemical methods. The selfcatalyzed tribo-electrochemistry strategy will open a new thoroughfare for the protection of high-energy-density alkali metal anodes. Nevertheless, our construction of K@MES has great advantages in some aspects compared to CCP, which also indicates the validity of our work. In addition, based on the reviewer suggestion, we also introduce ref. 23 in more detail in the manuscript.

We have added the test results to the manuscript:

Revision: “The rupture of the SEI exacerbates the growth of dendrites²⁵, which changes the electric field distribution and ion flux. Researchers have developed a self-catalyzed tribo-electrochemistry strategy to construct a continuous and compact protective layer on the K electrode surface. This continuous and compact protective layer can not only improve K⁺ diffusion dynamics, but also inhibits K dendrite formation by increasing K⁺ conductivity and decreasing electron conductivity with the amorphous KF. However, there are some problems in cycle life and scale preparation.”

25. C. Qin *et al.*, Tribo-electrochemistry induced artificial solid electrolyte interface by self-catalysis. *Nat. Commun.* **12**, 7184 (2021).

3. The authors have shown its applicability to Li metal; would MES work for Na metal?

Response: Thank you very much for this valuable suggestion. The universality of

MES as an artificial and in situ metal electrode protection layer on alkali metal electrodes is also an extremely important aspect. We constructed three sodium metal anodes of Na@MES, Na@GO, and bare Na and assembled them as symmetric cells according to the reviewer's suggestion. The long cycle performance of three symmetrical batteries was tested and observed that MES plays a significant role in inhibiting dendrite growth and improving the cycle life of the symmetric cells. In addition, we add the test results of the sodium metal cells to the supplementary materials to improve the integrity of the whole work.

Symmetric cells were assembled using Na metal treated with different conditions and subjected to a long-cycle testing. Under the test conditions of a current density of 0.1 mA cm^{-1} and a deposition capacity of 0.1 mAh cm^{-1} (Figure R8), the plating/stripping lifetime of Na@MES is about 1200 hours (0.8M NaPF₆). However, the batteries with Na@GO and bare Na anodes showed a poor cycling life. The symmetric battery composed of Na@GO failed after cycling for about 550 hours, while the symmetric battery with bare Na could only cycle normally for 185 hours. This is due to the continuous growth of dendrites at the Na metal interface with a dramatic increase in interfacial instability, leading to a rapid short-circuiting of symmetric cells. A different electrolyte was used for the test (3M NaFSI in DME), the Na@MES symmetrical cell is still superior to other two Na anodes (Figure R9). It is worth noting that different electrolyte systems can show better stability and longer cycle life, which also proves the universality of MES.

Figure R8. Galvanostatic Na plating/stripping voltage profiles for the Na|Na symmetric cells. Current density of 0.1 mA cm^{-2} and capacity of 0.1 mAh cm^{-2} with NaPF_6 electrolyte.

Figure R9. Galvanostatic Na plating/stripping voltage profiles for the Na|Na symmetric cells. Current density of 0.1 mA cm^{-2} and capacity of 0.1 mAh cm^{-2} with NaFSI electrolyte.

Revision: "In addition, MES is applied to sodium metal batteries in the same way.

Tested in either NaPF₆ (**Supplementary Fig. 31**) or NaFSI (**Supplementary Fig. 32**) electrolyte, symmetrical cells assembled by Na@MES demonstrated the longest cycle life. These tests not only demonstrate the excellent performance of MES in suppressing dendrites and stabilizing the interface, but also reveal the universality of MES in metal anode applications.”

4. There is a notation K@MES|Cu (around page 11), but I think Cu@MES||K may be correct. Please check it.

Response: Thank you for your careful review of our work. We double-checked this notation in the manuscript and determined that the description of this section should be Cu@MES||K. We have changed the description of this section in the manuscript. In addition, we have re-examined the entire manuscript carefully to avoid such problems again. Again, we thank the reviewers for their review of this study and the entire manuscript.

We greatly appreciate the constructive feedback, which has vastly improved the overall quality of the revised manuscript. We hope our detailed explanations are satisfactory, and this reviewer will recommend *Nature Communications* to accept our revised manuscript.

Reviewer #2 (Remarks to the Author):

In this manuscript, the authors develop a strategy for stabilizing the metal interface with a metal electrode skin (MES) mimicking human skin, which is novel and creative. The combined effect of the ex-situ construction of a graphene interface layer and the in-situ formation of an enhanced SEI to form a metal electrode skin to protect the metal electrode. The MES-protected metal electrode is capable of achieving long cycle lives for both symmetric and asymmetric cells. K|Cu@MES achieves over 1600 cycles, which is one of the best potassium metal asymmetric cells reported to date. The full cell also shows excellent performance and cycling stability compared to previously published papers. Overall, the bionic interface protection approach reported in this manuscript is innovative. The 'biomimetic' concept used in this work provides an important advance in the design of future metal anodes. Therefore, this paper can be accepted by Nature Commun. Some suggestions are as follows.

Response: We appreciate your recognition and positive comments on this work, as well as your critical review and feedback on this work. We have made detailed revisions based on the specific comments and suggestions to improve the quality of our manuscript. In general, we hope that the revised manuscript will meet the requirements. The specific responses are as follows.

Comment 1: According to the information provided in the manuscript, the main material of the MES is F-GO. The surface of the F-GO modified metal electrode or copper foil is smooth and has a high degree of flatness, does this affect the electrolyte

infiltration?

Response: Thanks to the reviewer for this question. As the reviewer mentioned, the main material of MES is F-GO, the surface of metal electrode or copper foil after F-GO modification is smooth with high flatness. To further verify whether F-GO enhances electrolyte wetting, electrolyte contact angle tests were performed for Cu@F-GO, Cu@GO, and Cu. The contact angle refers to the angle θ between the tangent line of the gas-liquid interface and the solid-liquid boundary made at the intersection of gas, liquid and solid three phases. The contact angle tests were performed using electrolytes of two kinds (3M KFSI in DME and 0.8M KPF₆ in EC/DMC, EC:DMC=1:1). It is well known that different material surfaces have different wettability, and there is a correlation between wettability and wettability speed. The better the wettability of the material, the smaller the contact angle of the electrolyte. The contact angle test can be used to measure the contact angle between the material surface and electrolyte, and the size of the contact angle can directly determine the wettability. The better the wettability, the lower the internal resistance of the battery. As shown in the Figure R10, Cu@MES show a smaller contact angle, which proves that MES can promote electrolyte wetting and reduce the internal resistance of the cell. The experimental results show that F-GO does not affect the infiltration of electrolyte while improving the surface flatness of electrode.

Figure R10. Contact angles of three materials in different electrolytes. (a) Contact Angle of Cu@F-GO, Cu@GO and Cu at 0.8M KPF₆. (b) Contact Angle of Cu@F-GO, Cu@GO and Cu at 3M KFSI.

Revision: “In addition, the contact angle tests were performed using two kinds of electrolytes (3M KFSI in DME and 0.8M KPF₆ in EC/DMC, EC:DMC=1:1). The results of the contact angle test indicate that the modified surface can enhance the wettability of the electrolyte (Supplementary Fig. 14a and b).”

Comment 2: In Figure 2, the authors used COMSOL simulations to investigate the effect of surface flatness on the potential distribution and K deposition morphology. In the AFM tests and simulations, the authors used Cu and Cu@MES as a comparison, how does this differ from K and K@MES?

Response: Thanks to the reviewers for this question. We used COMSOL simulations to investigate the effect of surface flatness on the potential distribution and K deposition morphology. In the AFM tests and simulations, we used Cu and Cu@MES

as a comparison. In this work, we used Cu and Cu@MES as comparison samples for AFM testing, mainly because the Cu foil surface and the bare potassium surface have similar surface flatness (as shown in the Figure R11), and using Cu foil for comparison testing will not affect the final conclusions. In addition, potassium metal is ductile and deformation may occur during the test to affect the test results. Cu@MES and K@MES have the same surface, so an AFM test using Cu@MES or K@MES will give exactly the same results. The main purpose of the AFM test is to demonstrate that the modification of the electrode interface by MES can greatly improve the surface flatness. The purpose of comsol simulation is to show the effect of flatness on surface electric field and ion distribution at the electrode interface. Surface electric field and ion distribution can have a very important effect on metal plating and dendrite growth. Comsol simulation showed that the MES modified electrode had excellent surface flatness (Figure R12). It can homogenize electric field and ion distribution on electrode surface, inhibit dendrite growth, and finally improve the safety of potassium metal battery. The use of Cu and Cu@MES for testing and comsol does not affect the final conclusions.

Figure R11. Surface morphology of different materials. (a, b) SEM image of copper foil surface. (c, d) SEM image of potassium foil surface.

Figure R12. AFM and comsol simulation. (a) AFM images of copper foil and F-GO. (b) Surface electric field and corresponding ion distribution of the pole piece electrode during initial K plating of copper foil. (c) AFM images of F-GO. (d) Surface electric field and corresponding ion distribution of the pole piece electrode during initial K plating of F-GO.

Comment 3: Can MES on metal electrodes or Cu foil surfaces maintain structural stability and surface flatness after several cycles?

Response: Thank you for your constructive question. In order to further study whether MES can maintain structural stability and surface flatness after many cycles, we disassembled the symmetrical cell with 50 cycles, and took out the electrodes after the cycle for SEM and AFM tests. Figure R13 shows the SEM image of K@MES after the cycle. As shown in Figure R13a, the K@MES surface after circulation has a high flatness (no change compared with the initial state). SEM and EDS mapping showed that K@MES, MES remained on the top of potassium metal after many cycles. These

evidences show that MES still maintain structural stability and surface flatness after several cycles (Figure R13b, c).

Figure R13. The morphology and structure of K@MES after 50 cycles. (a) SEM image of surface at K@MES. (b, c) SEM image and EDS mapping of cross profile at K@MES.

Revision: “In addition, the K@MES was taken out after circulation for SEM test. By observing the surface and cross section of the electrode and characterizing by EDS, it can be found that MES can be retained on the surface of K metal in the state of stripping (Supplementary Fig. 36a-d) and plating (Supplementary Fig. 37a-d). These results indicate that MES can continuously protect metal electrodes. These results can also prove that MES can maintain structural stability and surface flatness after several cycles.”

Comment 4: In the Figure 3c, both K@MES and K@GO have artificial interfacial protection layers. Why do K@MES and K@GO assembled symmetrical cells have a lower impedance? How does the impedance of the different types of symmetrical cells

change during cycling?

Response: Thank the reviewer for raising this question. As shown in Figure 3c in the manuscript, a symmetric cell composed of K@MES has the lowest impedance and a symmetric cell composed of bare potassium has the greatest impedance. This pattern was maintained after 50 cycles. The smaller impedance in the initial cycle is due to the different wettability of the electrolyte at different interfaces, which can be verified by the contact Angle test of the electrolyte on different material surfaces. The better the infiltration, the lower the internal resistance of the battery. The results of the electrolyte contact angle test show that MES has the best infiltrative effect on the electrolyte. Therefore, under the same other test conditions, the symmetric cell composed of K@MES exhibits the lowest impedance during the initial test. After 50 cycles, the symmetrical cell composed of K@MES still maintains the minimum impedance. This is due to the fact that K@MES has the most stable interface (both artificial and enhanced SEI formed in situ) during the cycle. K@GO surface cannot stable SEI is formed during the cycle, the GO protective layer on the metal surface improves the interfacial flatness and facilitates uniform plating of potassium. The fundamental reason that bare potassium has the largest impedance is that infinite volume change of potassium in the plating process and uneven plating leads to continuous rupture and recombination of SEI, resulting in thicker and more uneven SEI at the interface, which greatly increases the interface impedance. Therefore, whether in the initial state or after many cycles, the symmetrical battery composed of K@MES can maintain the minimum impedance.

As shown in Figure R14, when the impedance of symmetrical batteries with different cycles is tested, K@MES can still maintain a stable impedance. Due to the instability of the interface, the symmetrical batteries composed of the other two metal anodes display a large variation during the cycle.

Figure R14. Impedance measurement. Impedance of three symmetrical cells at different cycles.

Revision: “In addition, symmetrical batteries with different number of cycles were tested. A symmetrical cell composed of K@MES exhibits stable low impedance. While K@GO and K exhibit the large and unstable impedance values. This shows that stable SEI plays an important role in stabilizing the interface (**Supplementary Fig. 27**).”

Comment 5: Supplementary Fig. 13a and b shows the lattice parameters of CF_x (x = 1) modelled substrate. Please carefully explain in the manuscript how CF_x (x = 1) differs from F-GO in the manuscript. The authors need to give a rationale or reference in the text for using CF_x (x = 1) for their calculations.

Response: We thank the reviewers for this question. In order to determine whether the electroplating process of potassium can promote the breakage of carbon fluorine

bonds, we construct a model and perform calculations using density functional theory. In constructing the model, we used fluorinated graphene CF_x ($x = 1$) instead of F-GO. As shown in the figure, K, graphene fluoride, KF, and graphene unit cells were modeled as the original reactants and products. The calculations tell that the carbon-fluorine bond of K will automatically break with contact of fluorinated graphene, reduced to KF and graphene. The dynamic insertion of each K atom releases an energy of 3.87 eV. The above results prove that the whole process is a spontaneous reaction. Corresponding to the defects present in F-GO in this work, fewer charges are transferred from the graphite layer to the fluorine atomic layer, and the C-F bond is more easily broken. Compared with fluorinated graphene, the carbon-fluorine coulomb force of F-GO is weaker, and the carbon-fluorine bond is easier to break. This indicates that the carbon-fluorine bonds in F-GO break more easily and the whole reaction process proceeds spontaneously. In addition, the initial material calculated using fluorinated graphene CF_x ($x = 1$) enables a more accurate determination of the Gibbs free energy change, which facilitates the understanding of the formation process of enhanced SEI (Figure R15).

Figure R15. Optimized structures of each modeling substrate. (a) In the models, the K, fluorine (F) and carbon (C) atoms are displayed as spheres in silver, yellow, and black, respectively. (b) The lattice parameter of K, graphene fluoride, potassium fluoride (KF) and graphene.

We sincerely appreciate this reviewer's constructive feedback, which has helped improve the quality of the revised manuscript. We hope our detailed explanations are

satisfactory, and this reviewer will recommend *Nature Communications* to accept our revised manuscript.

Reviewer #3 (Remarks to the Author):

This manuscript provided a metal electrode skin (MES) method for potassium metal battery (PMB) anodes. The perfectly designed MES can adjust the electric field and promote the in-situ formation of metal-fluoride nanocrystal-rich SEI layers. Interestingly, the ultra-long cycle life was achieved by Cu@MES|K asymmetric cells and K@MES|PB cells. I would like to recommend the publication of this manuscript after addressing the following issues.

Response: We are very grateful to the reviewer for the positive comments on this work, as well as for the critical review and feedback. We have made detailed revisions based on your specific comments and suggestions to improve the quality of our manuscript. In general, we hope that the revised manuscript will meet your requirements. The specific responses are as follows.

1. What's the difference between MES and artificial SEI? Is MES just a new name for artificial SEI?

Response: We thank the reviewer for this question. MES includes artificial SEI, but not only artificial SEI. Our research work is inspired by human skin. In the manuscript, we describe the structure and function of human skin. The skin is divided into two layers, the epidermis and the dermis. The epidermis is on the surface of the skin and it mainly functions as a barrier to protect against mechanical damage, physical damage, chemical damage, and microorganisms. The role of the dermis is to

make the skin more extensible and elastic. It protects the capillaries, glands and various substances in the dermal tissue. There are many ion channels in the dermis, which can be used as absorption channels for water and nutrients.

We construct a protective layer of metal electrodes similar to human skin. The first is the artificially added graphene protective layer, which is similar to the dermal layer in the skin with some elasticity. Part of the graphene layer can act as a host and inhibit the volume change during potassium metal plating. During cell cycling, the plating of potassium metal drives the breakage of carbon and fluorine bonds in the material, forming an inorganic-rich SEI. This enhanced SEI has an excellent mechanical strength and provides better protection for metal electrodes. The enhanced SEI is similar to the epidermal layer of human skin, which can better isolate the contact between the solid electrode and the electrolyte to act as a perfect barrier. The two protection mechanisms work together to form the MES on the metal surface, which has a superior protection ability for the metal electrode compared to the single artificial SEI (Figure R16).

Figure R16. The formation process of MES.

2. As the author says, ‘The skin is divided into two layers, the epidermis and the dermis.’ ‘The epidermis mainly displays a protection function and the dermis is to make the skin more extensible and elastic.’ The protection function of MES has been greatly reflected in the article, and I wonder whether MES is extensible and elastic. I suggest the author do some experiments about mechanical strength and metal fatigue to prove it.

Response: Thank you very much for this valuable suggestion. According to the reviewer's suggestion, we conducted mechanical strength and metal fatigue tests on MES, which further proved its extensible and elastic properties. The main material of MES is F-GO, and metal fatigue measurement is mainly aimed at F-GO. As shown in Figure R17, we use a simple way to test the metal fatigue performance of F-GO. Figures R17a and 17b show two schematic diagrams of folding in different directions. The metal fatigue properties of F-GO and GO are verified by folding tests on Cu@F-GO and Cu@GO. In the original state, both polar plates showed the good

integrity (Figure R17c). First of all, the folding experiment is carried out according to the first folding mode. After ten folds, F-GO still maintains structural stability, while cracks appear on GO surface (Figure R17d). After 30 folds, GO breaks but F-GO remains structurally stable (Figure R17e). By using another folding method for 30 times, the structure of F-GO did not change, but GO had a wide range of cracks (Figure R17f). These results indicate that F-GO has better metal fatigue performance. Benefiting from the increase of F content, the enhance SEI with high mechanical stability, which was confirmed by evaluating its modulus with different depth (5, 6 and 7 nm). As shown in Figure R18, with the increase of indentation depth, the modulus of SEI of F-GO surface increases sharply. At 5, 6, and 7 nm, the modulus of SEI are 5.3, 8.9 and 14.2 GPa, respectively. Correspondingly, the SEI on the GO surface has only 2.8, 3.6 and 4.5GPa at the same position, respectively. The results indicated that the increase of fluorine content greatly improved the mechanical strength of SEI, which played an important role in preventing SEI breakage and inhibiting dendrite generation.

Figure R17. Metal fatigue test for MES and GO. (a, b) Folding mode of metal fatigue experiment. (c) Cu@MES and Cu@GO in their original state. (d) Cu@MES and Cu@GO fold ten times (according to the method in Figure a). (e) Cu@MES and Cu@GO fold thirty times (according to the method in Figure a). (f) Cu@MES and Cu@GO fold thirty times (according to the method in Figure b).

Figure R18. SEI modulus for F-GO and GO surface after 50 cycles. (a) Schematic of the AFM tapping mode to test the modulus of SEI. The tapping depth increases stepwise, from 5, 6 to 7 nm. The 2D modulus mapping of each depth of SEI layers for (b) F-GO and (c) GO respectively.

Revision: “The repeated folding experiment shows that F-GO has a better metal fatigue performance (**Supplementary Fig. 9a-f**).”

“Benefiting from the increase of F content, the enhance SEI with high mechanical stability was confirmed by evaluating its modulus with different depth (5, 6 and 7 nm). As shown in **Supplementary Fig. 21**, with the increase of indentation depth, the modulus of SEI of F-GO surface increases sharply. At 5, 6, and 7 nm, the modulus of SEI are 5.3, 8.9 and 14.2 GPa, respectively. Correspondingly, the SEI on the GO surface has only 2.8, 3.6 and 4.5 GPa at the same position, respectively. The results indicated that the increase of fluorine content greatly improved the mechanical strength of SEI, which played an important role in preventing SEI breakage and inhibiting dendrite generation. The stable and dense SEI protects the electrode sheet from corrosion by the electrolyte (**Supplementary Fig. 22**). To demonstrate the protective effect of MES on the electrode, the potential holding test of the Cu@MES electrode at 4 V vs. K⁺/K was carried out. In the process of test, Cu@MES||K can keep 4 V, and current is zero, indicating that Cu is protected MES and electrolytes do not penetrate F-GO due to desolvation on the SEI, and K⁺ ions diffuse through the F-GO. As a comparison sample, the bare Cu failed to maintain a normal voltage of 4 V during the test, and the current in the battery gradually increased to the instrument's

maximum range. It indicates that the bare copper is seriously corroded.”

3. Potassium is a particularly active metal. PMB should have a great safety hazard in the application, so what is its application prospect?

Response: We appreciate the reviewers' attention and questions in this area. Indeed, potassium is a particularly active metal. However, PMB still have unique advantages and good application prospects. First of all, the advantages of potassium metal batteries are mainly reflected in the low cost of potassium and low chemical potential (-2.93 V vs. the standard hydrogen electrode, SHE). The potassium (K) element is abundant content in the earth's crust (~ 2.09 wt%, vs. ~ 0.0017 wt% of Li). (Tribo-electrochemistry induced artificial solid electrolyte interface by self-catalysis) Aluminum foil can replace copper foil as a current collector in potassium ion batteries (PIBs), which will not only significantly reduce the price of PIBs, but also reduce the weight of the current collector and increase the battery energy density. Potassium has the largest atomic radius (1.38 Å) compared with lithium (0.68 Å) and sodium (0.97 Å). In Propylene carbonate (PC) solvent, the Stoker radius of K^+ (3.6 Å) is smaller than that of Li^+ (4.8 Å) and Na^+ (4.6 Å), indicating that K^+ has the highest ion mobility and ion conductivity. In addition, molecular dynamics simulations (MDS) have demonstrated that the diffusion coefficient of K^+ is about 3 times that of Li^+ . These advantages make potassium ion batteries have high research and application value¹⁻³.

As a new secondary alkali metal battery, PMB can not only replace lithium metal

battery, but even replace some low-end lithium ion battery⁴. In addition, due to the lower cost of raw materials and relatively high energy density, potassium metal batteries are expected to be used in large-scale grid energy storage systems⁵. PMB research is still in its infancy, and we believe that with the continuous efforts of researchers, potassium ion batteries and PMB can be used more widely.

1. W. Zhang, Y. Liu, Z. Guo, Approaching high-performance potassium-ion batteries via advanced design strategies and engineering. *Sci. Adv.* **5**, eaav7412 (2019).
2. T. Hosaka, K. Kubota, A. S. Hameed, S. Komaba, Research Development on K-Ion Batteries. *Chem. Rev.* **120**, 6358-6466 (2020).
3. Y. Xu *et al.*, Highly nitrogen doped carbon nanofibers with superior rate capability and cyclability for potassium ion batteries. *Nat. Commun.* **9**, 1720 (2018).
4. J. Wang, W. Yan, J. Zhang, High area capacity and dendrite-free anode constructed by highly potassiophilic Pd/Cu current collector for low-temperature potassium metal battery. *Nano Energy* **96**, 107131 (2022).
5. J. Meng *et al.*, Amine-Wetting-Enabled Dendrite-Free Potassium Metal Anode. *ACS Nano*, (2022).

4. Based on question 3, I wonder whether the MES method can improve the safety of PMB, and suggest the authors do more supplement-relevant experiments.

Response: Thanks for the reviewer's suggestion. In the battery, MES on potassium metal is mainly to inhibit the growth of dendrites and prevent internal short circuit of the battery, so as to improve the safety of PMB. According to the suggestions of

reviewers, we conducted related supplementary experiments to further verify the protection of MES on potassium metal and improve the safety of PMB. As shown in Figure R19, SEM test was conducted on K@MES after circulation in order to further demonstrate the protection of MES on potassium metal. Figure R19a, b shows the surface and cross section after K@MES stripping 0.5mAh cm^{-2} , from which we can observe that the surface of K@MES in the stripped state is flat. MES is retained on the surface of potassium metal, which provides continuous protection to the metal electrode and inhibits the generation of dendrites. Figure R19c, d shows the surface and cross section after K@MES plating 0.5mAh cm^{-2} . The overall structure of the electrode is basically the same as that of the stripped electrode. MES retained on the surface of potassium metal can continuously inhibit the dendrites of the electrode and improve the safety of PMB.

In addition, to further demonstrate the protective effect of MES on potassium metal and PMB, we tested metal electrodes in a more aggressive manner. The recycled K@MES was removed from the PMB and placed in an air environment to test the heat distribution of the metal electrodes using an infrared thermal imager. As shown in Figure R20, at the initial stage, the bare potassium electrode reacted violently with the air, releasing a large amount of heat (0min), while the metal electrode protected by MES could isolate the air and slow down the reaction rate. With the increase of reaction time, the bare potassium electrode maintained the violent reaction with the air, showing a higher temperature. K@MES electrode remained stable (30min) after a long exposure to air, which fully demonstrated that MES can protect potassium metal

and improve the safety of PMB.

Figure R19. SEM image of K@MES surface and section after cycles. (a, b) surface and cross section at 0.5mAh cm^{-2} stripping, (c, d) surface and cross section at 0.5mAh cm^{-2} plating.

Figure R20. Potassium metal exothermic test. (a, b) Exothermic state of Bare K and K@MES after reaction with air.

Revision: “In addition, to further demonstrate the protective effect of MES on potassium metal and PMB, we tested metal electrodes in a more aggressive manner. The recycled K@MES was removed from the PMB and placed in an air environment to test the heat distribution of the metal electrodes using an infrared thermal imager. As shown in **Supplementary Fig. 41**, at the initial stage, the bare potassium electrode reacted violently with the air, releasing a large amount of heat (0min), while the metal electrode protected by MES could isolate the air and slow down the reaction rate. With the increase of reaction time, the bare potassium electrode maintained the violent reaction with the air, showing a higher temperature. K@MES electrode remained stable (30min) after a long exposure to air, which fully demonstrated that MES can protect potassium metal and improve the safety of PMB.”

5. In the section named ‘In situ formation of enhanced SEI’, It seems that the available evidence is insufficient for the in-situ variation of enhanced SEI. I suggest that the authors add some in-situ characterization to further illustrate the changes of SEI in charge-discharge cycles.

Response: Thanks for the reviewer suggestion. As the reviewer mentioned in question 3, potassium is a particularly active metal. In situ testing of potassium metal surface is very difficult and may not get correct results. However, we can basically achieve the in situ enhancement characterization of SEI through the ex situ XPS test. As shown in **Figure R21**, XPS spectra tests were performed on surface SEI with different plating

capacities. The composition and content of SEI were obtained by testing the surface of different materials by XPS. As shown in Figure R21a, in addition to the conventional sulfides, carbonates, nitrides and oxides, the contents of fluorides are clearly distinguished. With the increase of deposition area capacity, the content of F on MES surface increased gradually. This means that the more potassium is deposited, the more fluorine is released from F-GO. The mechanical strength of SEI can be improved effectively by increasing the inorganic content of SEI (Figure R21b).

In addition, the MES surface deposited at 0.5mAh cm^{-2} was etched at different depths before XPS tests (Figure R22a). Content of element F at different depths is basically the same, and is higher than that on the GO surface (Figure R22b). This indicates that the enhanced SEI has a uniform F distribution, and the increase of inorganic material benefits the mechanical strength of SEI. SEI enhancement also plays an important role in inhibiting dendrite growth and increasing cycle life of metal batteries.

Figure R21. The XPS spectra of MES after different area capacity with 3M KFSI

in DME as electrolyte. (a) Full survey XPS spectra. (b) High resolution F1s XPS spectra and fluoride content.

Figure R22. The XPS spectra of MES after different depth of etching with 3M KFSI in DME as electrolyte (0.5mAh cm^{-2}). (a) Full survey XPS spectra. (b) High resolution F1s XPS spectra and fluoride content.

Revision: “To further verify that K plating promotes the formation of enhanced SEI on the electrode surface, XPS spectra tests were performed on surface SEI with different plating capacities. The composition and content of SEI were obtained by testing the surface through XPS. As shown in **Supplementary Fig. 18a**, in addition to the conventional sulfides, carbonates, nitrides and oxides, the contents of fluorides are clearly distinguished. With the increase of deposition area capacity, the content of F on MES surface increased gradually. This means that the more potassium is deposited, the more fluorine is released from F-GO. The mechanical strength of SEI can be improved effectively by increasing the inorganic content of SEI (**Supplementary Fig. 18b**). In addition, the MES surface deposited at 0.5mAh cm^{-2} was etched at different depths and then XPS was performed (**Supplementary Fig. 19a**). Content of element F at different depths is basically the same, which is higher than that on the GO surface

(Supplementary Fig. 19b). This indicates that the enhanced SEI has a uniform F distribution, and the increase of inorganic matter benefits the mechanical strength of SEI.”

6. Some related refs should be cited, including Nano-Micro Lett. 2022, 14, 210 and Nature Communications, 2022, 13, 3209.

Response: We appreciate the reviewers' attention and questions in this area. More excellent literature would help us to improve the quality of the manuscript and make it easier for readers to understand the study. We have read these papers carefully and cited them in appropriate places according to the reviewers' suggestions.

Revision:

14. Z. Sun *et al.*, Expanding the active charge carriers of polymer electrolytes in lithium-based batteries using an anion-hosting cathode. *Nat. Commun.* **13**, 3209 (2022).
19. X. Li *et al.*, Quasi-Solid-State Ion-Conducting Arrays Composite Electrolytes with Fast Ion Transport Vertical-Aligned Interfaces for All-Weather Practical Lithium-Metal Batteries. *Nano-Micro Lett.* **14**, 210 (2022).
28. Z. Wu *et al.*, Deciphering and modulating energetics of solvation structure enables aggressive high-voltage chemistry of Li metal batteries. *Chem* **9**, 1-15 (2022).
30. W. Zhang, Y. Liu, Z. Guo, Approaching high-performance potassium-ion batteries via advanced design strategies and engineering. *Sci. Adv.* **5**, eaav7412

(2019).

We sincerely appreciate the constructive feedback, which has vastly improved integrality of this work and, thus, the overall quality of the revised manuscript. We hope our detailed explanations are satisfactory, and this reviewer will recommend *Nature Communications* to accept our revised manuscript.

REVIEWERS' COMMENTS

Reviewer #1 (Remarks to the Author):

The revised version of the manuscript adequately addresses the comments. I recommend accepting the manuscript in its current form.

Reviewer #2 (Remarks to the Author):

The authors have revised the paper. It can be accepted.

Reviewer #3 (Remarks to the Author):

Recommendation: accept

This manuscript provided a metal electrode skin (MES) method for potassium metal battery (PMB) anodes. The perfectly designed MES can adjust the electric field and promote the in-situ formation of metal-fluoride nanocrystal-rich SEI layers. Interestingly, the ultra-long cycle life was achieved by Cu@MES|K asymmetric cells and K@MES|PB cells. The authors have given convincing evidence according to the modification opinions. Therefore, I believe this paper can be accepted by Nature Communication.

Response to reviewer's comments:

Reviewer #1 (Remarks to the Author):

The revised version of the manuscript adequately addresses the comments. I recommend accepting the manuscript in its current form.

Response: We are very grateful to the reviewers for their evaluation and support.

Reviewer #2 (Remarks to the Author):

The authors have revised the paper. It can be accepted.

Response: Thank the reviewers for their suggestions on improving the quality of the work and their recognition of the work.

Reviewer #3 (Remarks to the Author):

This manuscript provided a metal electrode skin (MES) method for potassium metal battery (PMB) anodes. The perfectly designed MES can adjust the electric field and promote the in-situ formation of metal-fluoride nanocrystal-rich SEI layers. Interestingly, the ultra-long cycle life was achieved by Cu@MES|K asymmetric cells and K@MES|PB cells. The authors have given convincing evidence according to the modification opinions. Therefore, I believe this paper can be accepted by Nature Communication.

Response: We are very grateful to the reviewers for their support and recommendation of our work.